# Data Acquisition via Experimental Design for Data Markets

**Charles Lu**[*]
MIT

**Baihe Huang**
UC Berkeley

**Sai Praneeth Karimireddy**
USC, UC Berkeley

**Praneeth Vepakomma**
MBZUAI, MIT

**Michael I. Jordan**
UC Berkeley

**Ramesh Raskar**
MIT

## Abstract

The acquisition of training data is crucial for machine learning applications. Data markets can increase the supply of data, particularly in data-scarce domains such as healthcare, by incentivizing potential data providers to join the market. A major challenge for a data buyer in such a market is choosing the most valuable data points from a data seller. Unlike prior work in data valuation, which assumes centralized data access, we propose a federated approach to the data acquisition problem that is inspired by linear experimental design. Our proposed data acquisition method achieves lower prediction error without requiring labeled validation data and can be optimized in a fast and federated procedure. The key insight of our work is that a method that directly estimates the benefit of acquiring data for test set prediction is particularly compatible with a decentralized market setting.

## 1 Introduction

While massive training datasets enable major machine learning breakthroughs, they remain largely inaccessible outside of large companies, motivating mechanisms for broader data access. A related point is that many data owners have become resistant to having their data collected indiscriminately without their consent or without their participation in the fruits of predictive modeling, resulting in legal challenges against prominent AI companies [16, 33]. These trends motivate the study of *data marketplaces*, which aim to incentivize data sharing between sellers, that provide access to data, and buyers, that pay compensation for data access [11, 1, 54].

For practical data acquisition scenarios, a data buyer has a specific goal in mind and, in particular, wants training data to predict their test data in a specified context. Accessing different datapoints may require varying prices associated with each datapoint, which may reflect heterogeneous cost, quality, or privacy levels for each datapoint [44, 40].

For example, consider a hospital that wants to make a prediction for a specific patient's X-ray. The hospital can submit this X-ray as an unlabeled test query to the marketplace, along with a budget to access relevant training data. The marketplace selects useful training data to build a model for this specific prediction task, sharing only the final prediction rather than the raw data to protect privacy. This process can be repeated for each new patient query (see Figure 1). However, not all X-ray images will be equally relevant. Thus, we want to select only those seller datapoints that are most useful for answering the buyer's query and fit the buyer's budget.

This goal of *data acquisition* has motivated the development of many data-valuation techniques (e.g., [23, 29, 38, 60, 52, 39, 57, 46, 30]). However, we argue that current data valuation techniques

---

[*]luchar@mit.edu

38th Conference on Neural Information Processing Systems (NeurIPS 2024).

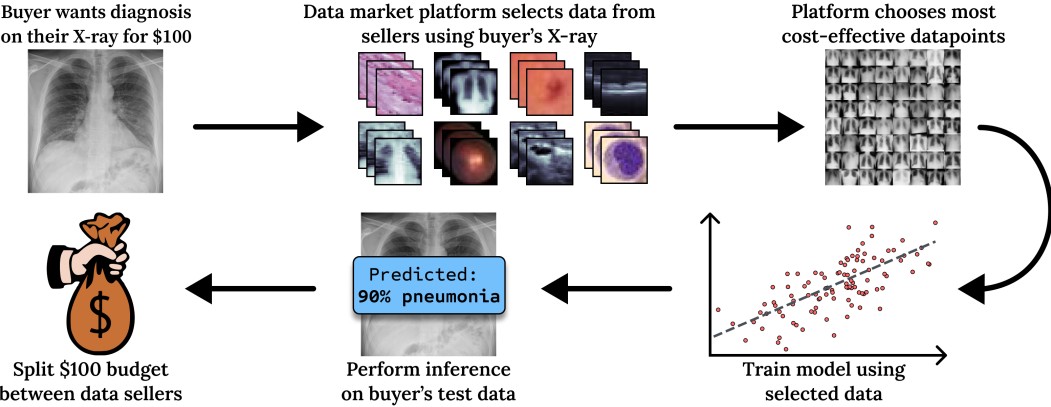

Figure 1: **Overview of Data Marketplace Approach.** A buyer brings their test query (e.g., a patient's chest X-ray needing diagnosis) and a budget to the marketplace. DAVED selects the most relevant subset of seller training data to minimize prediction error on the buyer's specified test query while respecting budget constraints. Unlike prior methods that require labeled validation data, DAVED directly optimizes for test performance. This enables targeted, cost-effective data acquisition compared to purchasing entire datasets.

are misaligned with the data acquisition problem, particularly in the context of data marketplaces. They all face at least one of the following limitations:

- The selection process may not be adaptive to the buyer's (unlabeled) test queries, potentially failing to identify the most relevant data. In a data marketplace, buyers typically need to purchase only a small subset of datapoints most relevant to their test data, which may follow a significantly different distribution than the overall seller data.

- When adaptive selection is implemented, these techniques rely on labeled validation data, which is often impractical. Further, when a small quantity of such data is available, the selection may overfit the validation data and result in poor performance on the test queries.

- The algorithms are not scalable and typically require retraining the ML model numerous times. Hence, they are unable to select from realistic seller corpora ($>$100K+ datapoints).

Instead, we propose **data acquisition via experimental design** (DAVED) method that overcomes all of these limitations. Unlike most previous work in data valuation, our approach does not require a labeled validation dataset and instead directly optimizes data selection for the buyer's unlabeled test queries.

Additionally, our approach accounts for budget constraints and is able to weigh the price of each seller's datapoint against its potential benefit, simultaneously solving the budget and revenue allocation problems [64]. Moreover, it can be implemented in a federated manner, achieving lower prediction error even compared to centralized baselines.

Our contributions are the following:

1. Formulate the data acquisition problem for data marketplaces and demonstrate that data valuation methods make a fundamental theoretical mistake of "inference after selection" (Theorem 1).

2. Design a novel, highly scalable, and distributed data selection procedure that eliminates the need for validation data by directly selecting the most cost-effective seller data to answer the buyer's test queries (Algorithm 1).

3. Demonstrate state-of-the-art performance on synthetic data and medical data in various modalities (X-rays, images, and text).

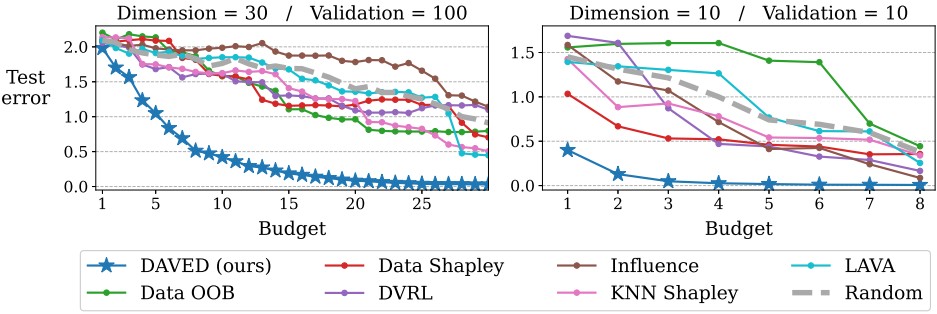

Figure 2: **Failures of Current Validation-based Data Valuation Methods.** Current data valuation methods overfit when data dimensionality is high or validation sets are small. Using 1,000 seller datapoints (each with cost 1) of Gaussian distributed data, we compare test error across methods as buyers acquire data under different budgets. (Left) Validation-based data valuation methods overfit when the data is too high dimensional ($d = 30$). (Right) Even with low-dimensional data ($d = 10$), overfitting occurs when the validation set is too small ($n = 10$), resulting in worse performance than random selection. Our validation-free method (DAVED) method maintains low error in both scenarios.

## 2    Data Acquisition versus Data Valuation

In a decentralized data marketplace, data acquisition must be performed *before* full data access is granted to the buyer [36]. This relates to Arrow's Information Paradox [7] — sellers are unwilling to share data before payment, while buyers need to evaluate utility before purchasing. This distinction between data valuation and data acquisition for data marketplaces is also discussed in a recent data acquisition benchmark, where data value must be estimated without requiring white-box access to the seller's data (i.e., full, unrestricted access to the data) [12].

A more fundamental issue with validation-based data valuation approaches is exemplified by the Data Shapley value approach [23, 29, 38, etc.], which measures the marginal contribution of each training datapoint's improvement to a validation metric. They are great for after-the-fact attributing the relative influence of the training data. However, they cannot be used to *make decisions* about which datapoints should be included in the training. This is because of, as noted earlier, the *"inference after selection"* issue. Using validation data to select training data leads to substantial over-fitting to the validation data.

*Illustrative example:* Suppose that we only have a single validation datapoint. Then, it is clear that we will select training data similar to this singular datapoint, and our selection has no hope of working on the test dataset. While this clearly demonstrates overfitting in an extreme scenario, we show in Figure 2 that increasing the validation set size does not circumvent this issue. We see that other data valuation techniques have poor test prediction errors—some techniques even underperform even a random selection baseline! This clearly demonstrates overfitting. Our proposed method maintains low test error as more seller training data is selected. In Section 3, we will dig in deeper into this phenomenon and prove a very strong theoretical lower bound. We show that any approach that relies upon validation data for data selection can perform as badly as *throwing away all the training data* and simply training on the validation data alone! This is especially true when our budget is small compared to the dimensionality of the problem, as is likely in a data market setting — the data is typically high dimensional, and we can only select a very small fraction of the total available data.

## 3    Setup and Limitations of Prior Methods

**Description of Data Acquisition Setting.** As shown in Figure 1, in our setting of data acquisition, a buyer has a budget and test data. The platform uses the buyer's data to select training datapoints from the seller that optimize buyer test error. The interaction proceeds as follows:

1. The buyer brings their test data $X^{\text{test}} = [x_1^{\text{test}}, \ldots, x_m^{\text{test}}]$ and a budget $B$ to the market platform. This test data is associated with unknown target labels $Y^{\text{test}} = [y_1^{\text{test}}, \ldots, y_m^{\text{test}}]$.

2. The platform also has access to $n$ datapoints from data sellers $Z^{\mathrm{train}} = \{(x_j, y_j)\}_{j=1,\dots,n}$ and their associated costs $\{c_j\}_{j=1,\dots,n}$.

3. Given *only the covariates* $X^{\mathrm{test}}$, the platform assigns a selection weight $w_j$ to each datapoint $(x_j, y_j)$. This weight $w_j \in \{0, 1\}$ represents the discrete action of selecting datapoint $j$.

4. The platform selects datapoints from the sellers according to $\mathbf{w} = (w_1, \dots, w_n)$, trains a model $f_{\hat{\theta}(\mathbf{w})}$, makes the predictions $f_{\hat{\theta}(\mathbf{w})}(X^{\mathrm{test}})$, and distributes $c_j \cdot w_j$ payment for each datapoint used in training.

In general, we do not make i.i.d assumptions between the train and test - we expect the test queries will not be similar to the total available train data. The goal then is to pick the weights $\mathbf{w}$, which minimizes the prediction error for the buyer while adhering to their budget constraint $\sum_{j=1}^{n} w_j c_j \leq B$. This gives rise to the following problem:

$$\min_{\mathbf{w} \in \{0,1\}^n} \mathcal{L}(\mathbf{w}) := \frac{1}{m} \sum_{i=1}^{m} \mathbb{E}\left[ l\left( f_{\hat{\theta}(\mathbf{w})}(x_i^{\mathrm{test}}), y_i^{\mathrm{test}} \right) \right] \quad \text{s.t.} \sum_{j=1}^{n} w_j c_j \leq B, \tag{1}$$

where $l$ is squared loss. Here, the expectation is over the conditional label distribution of $y_i^{\mathrm{test}} | x_i^{\mathrm{test}}$, and the potential randomness of the algorithm. Note that this problem can not be solved because we do not know the targets $Y^{\mathrm{test}}$. Instead, we need to rely on a surrogate objective function (proxy) $\hat{\mathcal{L}}(\mathbf{w})$. One approach to constructing such a proxy is by using validation data.

**Folly of Relying on Validation Data.** In most data valuation methods, e.g., Data Shapley [23], the value of data is evaluated using a labeled validation set $Z^{\mathrm{val}} = \{(x_j^{\mathrm{val}}, y_j^{\mathrm{val}})\}_{j=1}^{n_{\mathrm{val}}}$. Implicitly, these methods assume that the known $Z^{\mathrm{val}}$ is drawn from the same distribution as the unknown $Z^{\mathrm{test}}$. Then using this validation data, scores $(s_1, \dots, s_n)$ are assigned to the seller training datapoints $Z^{\mathrm{train}}$. For two datapoints $i, j \in \mathrm{train}$, the score $s_i > s_j$ if the datapoint $i$ is more valuable than $j$ [46, 30]. More concretely, $s_i > s_j$ implies that training with $i$ would lead to a smaller validation loss than if $j$ was used instead. Thus, these scores can used to select the most valuable datapoints.

However, note that we used the validation dataset to compute the scores. Thus, selecting the top-k scores results in implicitly minimizing the validation loss i.e., all validation-based data valuation schemes implicitly optimize the following proxy loss

$$\min_{\mathbf{w} \in \{0,1\}^n} \hat{\mathcal{L}}^{\mathrm{val}}(\mathbf{w}) := \sum_{j=1}^{n_{\mathrm{val}}} l\left( f_{\hat{\theta}(w)}(x_j^{\mathrm{val}}), y_j^{\mathrm{val}} \right) \quad \text{s.t.} \sum_{j=1}^{n} w_j c_j \leq B. \tag{2}$$

This approach heavily relies on the quantity and quality of the validation dataset in order to generalize to the actual test dataset. In fact, we have the following minimax lower bound even when restricting ourselves to simple linear models.

---

**Theorem 1** (Informal version of Theorem A.1). *Let $\mathbf{w}^*$ denote the solution of our original problem* (1) *and $\hat{\mathbf{w}}$ solve* (2). *Suppose that all our data $Z^{\mathrm{train}}, Z^{\mathrm{val}}, Z^{\mathrm{test}}$ are drawn i.i.d. from some distribution $\mathcal{D}_{X,Y}$ where $\mathcal{D}_X$ is supported on $B_R^d$ (zero-centered ball with radius $R$ in $\mathbb{R}^d$), and $Y = \theta^\top X + \varepsilon$ where $\varepsilon$ is independent zero-meaned noise with variance $\sigma^2$. For any training algorithm, when the number of training data is sufficiently large, with high probability,*

$$\inf_{\hat{\theta}} \sup_{\mathcal{D}_{Y|X}} \mathbb{E}_{X^{\mathrm{test}}} \left[ \mathcal{L}(\hat{\mathbf{w}}) - \mathcal{L}(\mathbf{w}^*) \right] \gtrsim \frac{\sigma^2 d}{n_{\mathrm{val}}}.$$

---

This result implies that the expected test error for any validation-based approach can in the worst-case scale as $d/n_{\mathrm{val}}$ with high probability. This dependence on the dimension $d$ and the number of validation points $n_{\mathrm{val}}$ highlights that this method may be suboptimal in high-dimensional settings or when the validation dataset is small. In fact, we would get the same error scaling if we threw away the training data and trained a model $\hat{\theta}$ on the $n_{\mathrm{val}}$ validation datapoints alone. This explains the striking overfitting we observed earlier in Figure 2. Furthermore, obtaining a large amount of ground-truth labeled data may be challenging in many real-world applications. Instead, we propose a validation-free approach to data acquisition based on experimental design.

# 4 Our Methods and Implementations

We propose an alternative approach based on a proxy objective. Our key assumption is that the conditional distribution $\mathcal{D}_{y|x}$: $y = f_{\theta_*}(x) + \epsilon$ is identical across $Z^{\text{train}}$ and $Z^{\text{test}}$. This is a natural assumption in many domains — for instance, if an X-ray exhibits indicators of a specific disease, it should receive the same diagnosis regardless of whether it appears in the training or test set. Without this assumption, our problem becomes intractable since the same $x_j$ could map to arbitrarily different labels across train and test sets. Under this framework, we can reformulate our problem using V-optimal experiment design [47].

**Step 1: Linearizing the problem.** Our goal is to design a proxy loss function $\hat{\mathcal{L}}(\mathbf{w})$ which approximates the true test loss $\mathcal{L}(\mathbf{w})$. To do this, we have to reason about how different choices of training data $S \subset Z^{\text{train}}$ could impact the prediction on a particular test datapoint in $X^{\text{test}}$. This is a notoriously challenging problem for general deep learning models [9]. Instead, we use a linear approximation and model the complicated training dynamics with kernelized linear regression. We suppose we have a known feature-extractor $\phi : \mathcal{X} \to \mathbb{R}^{d_0}$ and an unknown $\theta^* \in \mathbb{R}^{d_0}$ such that the data is generated as

$$y = \theta^{*\top} \phi(x) + \varepsilon, \tag{3}$$

where $\varepsilon$ is independent noise with mean zero and $d_0$ is the embedding dimensionality. The function $\phi(\cdot)$ can be the empirical Neural Tangent Kernel (eNTK) [27, 41, 58] of the model, or even the embeddings extracted from a deep neural network such as CLIP [49]. While this may be a bad approximation in general [61], a recent line of work has shown that such eNTK representation very closely approximates the *fine-tuning* dynamics of pre-trained models both theoretically [58, 42] as well as emperically [22, 63]. In fact, such linear approximations have also been used to speed up validation-based data attribution computations [46].

**Step 2: Experimental design proxy.** Given the assumption on our data from Eqn. (3), we can use the V-optimal experiment design framework [51, 47, 26] to define a proxy objective. First, suppose that $S \subseteq (\phi(X^{\text{train}}), Y^{\text{train}})$ is the subset selected by $\mathbf{w}$ and then we performed least-squares regression. The resulting estimate $\hat{\theta}(\mathbf{w})$ can be computed in closed form as

$$\hat{\theta}(\mathbf{w}) = \left( \sum_{j=1}^{n} w_j \phi(x_j) \phi(x_j)^\top \right)^\dagger \left( \sum_{j=1}^{n} w_j \phi(x_j) y_j \right).$$

Henceforth, we will drop the $\phi$ when obvious from context and simply use $x$. We can further use Eqn. (3) to compute the expected error on an arbitrary test query $x_0, y_0$ as follows:

$$
\begin{aligned}
\mathbb{E}[(\hat{\theta}(\mathbf{w})^\top x_0 - y)^2 | X^{\text{train}}, x_0] &\overset{a_1}{=} \mathbb{E}[((\hat{\theta}(\mathbf{w}) - \theta^*)^\top x_0 + \varepsilon)^2] \\
&\overset{a_2}{=} x_0^\top \mathbb{E}[(\hat{\theta}(\mathbf{w}) - \theta^*)(\hat{\theta}(\mathbf{w}) - \theta^*)^\top] x_0 + \mathbb{E}\|\varepsilon\|^2 \\
&\overset{a_3}{=} x_0^\top \mathbb{E}[\hat{\theta}(\mathbf{w}) \hat{\theta}(\mathbf{w})^\top] x_0 + \mathbb{E}\|\varepsilon\|^2 \\
&\overset{a_4}{=} x_0^\top \big( \underbrace{\sum_{j=1}^{n} w_j x_j x_j^\top}_{=: \mathcal{I}(\mathbf{w})} \big)^\dagger x_0 + \mathbb{E}\|\varepsilon\|^2
\end{aligned}
$$

Here $a_3$ uses the unbiasedness of the ordinary least squares (OLS) estimator and $a_4$ plugs in the closed form of $\hat{\theta}(\mathbf{w})$ and simplifies. With this, we end up with a very clean expression for the expected test error on an arbitrary point $x_0$, and the matrix $\mathcal{I}(\mathbf{w})$ is known as the *Fisher information matrix*. While regression suffices for our use case, the procedure can be extended to general linear models. Dropping the fixed $\mathbb{E}\|\varepsilon\|^2$, we can use this to build our proxy function $\hat{\mathcal{L}}^{ED}(\mathbf{w})$ and arrive at the following optimization problem

$$\min_{\mathbf{w} \in \{0,1\}^n} \left\{ \hat{\mathcal{L}}^{ED}(\mathbf{w}) := {}^1\!/_m \sum_{i=1}^{m} (x_i^{\text{test}})^\top \mathcal{I}(\mathbf{w})^\dagger (x_i^{\text{test}}) \right\} \quad \text{s.t. } \sum_{j=1}^{n} w_j c_j \leq B. \tag{4}$$

This optimization objective directly measures how useful each training point would be for predicting the specified test query. The matrix $\mathcal{I}(\mathbf{w})$ captures how much information each selected datapoint provides about the test point in the embedded feature space.

Note that our proxy function $\hat{\mathcal{L}}^{ED}(\mathbf{w})$ can be computed using just $X^{\text{train}}, X^{\text{test}}$ and does not even need access to training labels. Unfortunately, the objective in (4) is NP-hard to optimize [4]. We next see how to derive fast and provably good approximation algorithms for (4).

**Step 3: Fast approximation.** To make Eq. 4 amenable to gradient-based optimization, we drop the constraint that $w_j \in \{0,1\}$ and allow it to be a continuous positive vector i.e., $\mathbf{w} \geq 0$ and $\sum_{j=1}^{n} w_j c_j \leq B$. With this relaxation, the proxy objective $\hat{\mathcal{L}}^{ED}(\mathbf{w})$ is continuous and convex in $\mathbf{w}$ [10]. We then run the "herding" variant of the *Frank-Wolfe* algorithm [59, 37, 53, 8, 65]. To do this, define $(\tilde{\mathbf{w}}_t := \mathbf{w}_t/\mathbf{c})$ for any $t$. We start from a $\tilde{\mathbf{w}}_0 = \mathbf{e}_0$ and iteratively update as[2]

$$\tilde{\mathbf{w}}_{t+1} \leftarrow (1 - \alpha_t)\tilde{\mathbf{w}}_t + \alpha_t \mathbf{e}_{j_t}, \quad \text{where } j_t = \arg\max_{j \in [n]}(-\nabla_{w_j}\hat{\mathcal{L}}(\mathbf{w}_t)/c_j) \tag{5}$$

Note that if we use the step-size $\alpha_t = \frac{1}{t+1}$ in (5), $\mathbf{w}_t$ satisfies a special property at any iteration $t$:

$$\tilde{\mathbf{w}}_t \in \Delta^n \text{ and further } (t+1)\tilde{\mathbf{w}}_t \in \{0,1\}^n .$$

Run the procedure until the last iteration $t = t_o$ for which we still have $\|\mathbf{w}_{t_0}\|_1 \leq B$. We can adapt the theory from [8, 28] to analyze the above procedure and show the following.

---

**Theorem 2** (Informal). *Let us run Frank-Wolfe herding update* (5) *for $t_0$ steps such that it is last step which satisfies $\|\mathbf{w}_{t_0}\|_1 \leq B$. We use $\tilde{\mathbf{w}}_{t_0} = ((t_0 + 1)\mathbf{w}_{t_0}/\mathbf{c})$ as our selection vector and we would have selected $t_0$ datapoints. Then, under some assumptions, we have*

$$\hat{\mathcal{L}}^{ED}\big((t+1)\tilde{\mathbf{w}}_{t_0}\big) \leq \min_{\mathbf{w} \in \{0,1\}^n, \sum_{j=1}^{n} w_j c_j \leq B} \hat{\mathcal{L}}^{ED}(\mathbf{w}) + O\left(\frac{\log t_0}{t_0}\right).$$

---

The above theorem shows that our continuous relaxation does not significantly affect the optimality of our result — we get $O(\frac{\log t_0}{t_0})$ close to the optimal solution to the original NP-hard (4). If all datapoints have equal cost $c$, then $t_0 = \lfloor B/c \rfloor$, and so our approximation quality improves as we increase the budget. While better approximation guarantees are attainable [3], their procedure is significantly more involved and is not easily amenable to efficient federated implementations as ours is.

**Step 4: Efficient federated implementation.** Our practical implementation directly restricts $\mathbf{w} \in \Delta^n$ instead of $\tilde{\mathbf{w}}$ in the theoretical implementation above i.e., we run

$$\mathbf{w}_{t+1} \leftarrow (1 - \alpha_t)\mathbf{w}_t + \alpha_t \mathbf{e}_{j_t}, \quad \text{where } j_t = \arg\max_{j \in [n]}(-\nabla_{w_j}\hat{\mathcal{L}}(\mathbf{w}_t)/c_j) \tag{6}$$

This way $\mathbf{w}$ can be directly interpreted to be the sampling probability for different seller training datapoints. The bottleneck to efficiently implementing (6) is computing the gradient. At step $t$, the negative gradient can be shown to be

$$g_j := -\nabla_{w_j}\hat{\mathcal{L}}(\mathbf{w}_t) = 1/m \sum_{i=1}^{m}\left((x_i^{\text{test}})^\top \mathcal{I}(\mathbf{w}_t)^\dagger(x_j^{\text{train}})\right)^2 . \tag{7}$$

Thus, if we have the inverse information matrix $\mathcal{I}(\mathbf{w}_t)^\dagger$ pre-computed, $g_j$ as well as the update (6) can be trivially computed by seller $j$ using only their data $x_j^{\text{train}}$ (and the test data). Next, we show how to efficiently maintain the inverse information matrix. Note that the update (6) has a special structure: all coordinates are shrunk, and then only a single coordinate of $\mathbf{w}_t$ is increased. We can relate the resulting $\mathcal{I}$ matrices with a rank-one update as:

$$\mathcal{I}(\mathbf{w}_{t+1}) = (1 - \alpha_t)\mathcal{I}(\mathbf{w}_t) + \alpha_t x_{j_t} x_{j_t}^\top .$$

Define $P_t := \mathcal{I}(\mathbf{w}_t)^\dagger$. We can use the Sherman–Morrison formula [50] to compute $P_{t+1} = \mathcal{I}(\mathbf{w}_{t+1})^\dagger$ as

$$P_{t+1} = \frac{1}{1 - \alpha_t}P_t - \frac{\alpha_t P_t x_{j_t} x_{j_t}^\top P_t}{1 - \alpha_t + \alpha_t x_{j_t}^\top P_t x_{j_t}} . \tag{8}$$

For each round $t$, this update only involves the current matrix $P_t = \mathcal{I}(\mathbf{w}_t)^\dagger$ and the single datapoint $x_{j_t}$ selected for the round. Thus, the seller can also locally compute this update as well as the updated cost $\hat{\mathcal{L}}(\mathbf{w})$ as in Eq. (4) for any $\alpha_t$:

$$\hat{\mathcal{L}}(\mathbf{w}) = \frac{1}{m(1-\alpha_t)} \sum_{i=1}^{m}\left(x_i^\top P_t x_i\right) - \frac{\alpha_t}{1 + \alpha_t x_{j_t}^\top P_t x_{j_t}} \sum_{i=1}^{m}\left(x_i^\top P_t x_{j_t}\right)^2 \tag{9}$$

---

[2]Here, $\mathbf{e}_j$ is the standard basis vector along axis $j$, and in $\tilde{\mathbf{w}} := \mathbf{w}/\mathbf{c}$, the division is performed element-wise.

**Algorithm 1** DAVED: Iterative Optimization Procedure

---

1: **Input:** buyer test datapoint $X^{\text{test}} \in \mathbb{R}^{m \times d}$, seller training data $X \in \mathbb{R}^{n \times d}$, seller weights $\mathbf{w} \in \Delta^n$, iteration steps $T$, regularization parameter $\lambda_{\text{Reg}} \in [0,1]$, and seller datapoint costs $\mathbf{c} \in \mathbb{R}^n_+$

2: $\mathbf{w}_0 \leftarrow \mathbb{1}/n$             # Initialize weight vector to uniform distribution

3: $P_0 \leftarrow \left((1 - \lambda_{\text{Reg}})\, X^\top \operatorname{diag}(\mathbf{w}_0)\, X + \lambda_{\text{Reg}} \cdot \sigma_X I_{n \times n}\right)^{-1}$      # Initialize $P$ (Eq. 10)

4: **for** $t \in \{1, 2, \dots, T\}$ **do**

5:     $g \leftarrow -\nabla \hat{\mathcal{L}}(\mathbf{w}_t)$             # Compute negative gradients (Eq. 7)

6:     $j_t \leftarrow \arg\max_j \; (g_j/c_j)$             # Select coordinate based on costs

7:     $\alpha_t \leftarrow \texttt{LINE\_SEARCH}(\hat{\mathcal{L}})$             # Find optimal step size (Eq. 9)

8:     $\mathbf{w}_{t+1} \leftarrow (1 - \alpha_t)\mathbf{w}_t + \alpha_t \mathbf{e}_{j_t}$    # Shrink weights and upweight the chosen coordinate (Eq. 6)

9:     $P_{t+1} \leftarrow \texttt{SHERMAN\_MORRISON}(P_t, x_{j_t}, \alpha_t)$       # Update inverse information matrix (Eq. 8)

10: **end for**

11: **Output:** Sample seller data according to $\mathbf{w}_T \in \Delta^n$ without replacement until budget $B$ runs out.

---

Thus, a line search can be performed to determine the optimal step size $\alpha_t \in [0,1]$ to minimize the proxy loss as. This differs from (5) where we used a specific choice of $\alpha_t$. Frank-Wolfe is known to be more stable with the line search [53, 65]. The seller can communicate this $\alpha_t$ and $x_{j_t}$ to the platform to compute the updated $P_{t+1}$ using only $O(d)$ communication.

An additional practical consideration is that by initializing $\mathbf{w_0} = c_1 \mathbf{e}_1$, we have an ill-conditioned inverse information matrix $P_0$. We instead use an initialization of $\mathbf{w_0} = \mathbb{1}_n/n \in \Delta^n$ and further add a feature-wise regularization term. This makes the initial $P_0$

$$P_0 = \left((1 - \lambda_{\text{Reg}})\, X^\top \operatorname{diag}(\mathbf{w_0}) X + \lambda_{\text{Reg}} \cdot \operatorname{diag}(\hat{\sigma})\right)^{-1}, \tag{10}$$

where $\hat{\sigma}_i = \sqrt{\frac{1}{n} \sum_{j=1}^n (X_{ji} - \bar{X}_i)^2}$ is the empirical standard deviation of feature $i$. The complete details are summarized in Algorithm 1.

**Single-step variant.** We can also forgo the iterative process and instead linearly approximate the cost function (Eq 4) with a *single step* that selects the top $k$ datapoints under the budget $B$,

$$\texttt{single\_step}(x^{\text{test}}, X, B) = \texttt{top\_k}\left(\Big\{ \sum_{i=1}^m \big[(x_i^{\text{test}})^\top P_0 x_j\big]^2 \Big\}_{j=1}^n\right). \tag{11}$$

This simplified version is extremely fast while still maintaining relatively good performance.

# 5 Experiments

We evaluate our proposed method for data acquisition (DAVED) against common data valuation methods on both synthetic data and four real-world medical:

1. **Fitzpatrick17K** [24], a skin lesion dataset, where the task is to predict Fitzpatrick skin tone on a 6-point scale from dermatology images.

2. **RSNA Pediatric Bone Age dataset** [25], where the task is to assess bone age (in months) from X-ray images of an infant's hand.

3. **Medical Information Mart for Intensive Care (MIMIC-III)** [31], where the task is to predict the length of hospital stay from 48 attributes such as demographics, insurance, and medical conditions.

4. **DrugLib reviews** [34], text reviews of drugs where the task is to predict ratings (1-10).

For validation-based methods, we use a validation set of 100 datapoints. We report mean test errors over 100 buyers. For more details on the experimental setup, see Appendix C. Our code is available at this repo: `https://github.com/clu5/data-acquisition-via-experimental-design`.

**Comparing Performance on Data with Homogeneous Costs.** In Figure 3, we evaluate our method and several other data valuation methods on varying amounts of Gaussian data with homogeneous

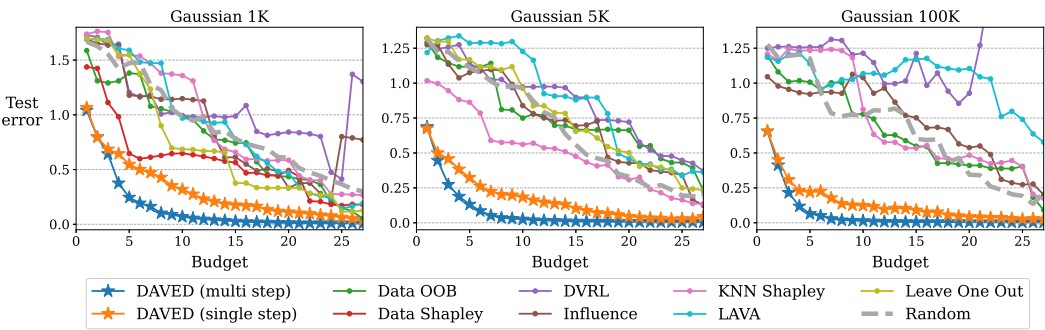

Figure 3: **Data Acquisition Performance across different Market Sizes on Synthetic Data.** We compare test prediction error as seller training data is selected under varying budgets and amount of data for sale, with total available seller data of 1K (left), 5K (middle), and 100K (right) points. Our data selection method (DAVED) consistently achieves lower MSE with fewer purchased datapoints, i.e., better data acquisition efficiency, than other data valuation methods. Both multi-step and single-step variants of DAVED achieve lower test MSE with fewer training points compared to validation-based methods. The performance gap is especially pronounced with small budgets (5-10 points). Unless otherwise specified, all results are averaged over 100 random test points.

fixed costs. Compared to other methods, both multi- and single-step versions of DAVED have lower test errors across budgets on synthetic data. This performance gap is especially large when the buyer has a small budget (around 5-10 seller training datapoints). In Figure 4, we evaluate our method on real image and text data embedded through CLIP and GPT-2 feature representations. We observe that DAVED has better performance compared to most other baselines on all three datasets, highlighting that the proposed method is practical for embeddings of high-dimensional data. Table 1 summarizes our results on all datasets. For the Gaussian data and MIMIC datasets, we report the mean error of budgets from 1 to 10, while for the embedded datasets (RSNA, Fitzpatrick17K, DrugLib), we report the mean error of budgets from 1 to 100 in intervals of five.

**Comparing Performance on Data with Heterogeneous Costs.** Next, we compare methods on seller data with non-homogeneous costs. We uniformly sample costs $c \in \{1, 2, 3, 4, 5\}$ for each seller datapoint and consider two cost functions, $c_j = \sqrt{c}$ and $c_j = c^2$, which downweights gradient of that datapoint $x_j$ (see Equation 7). To simulate heterogeneous utility across datapoints, we introduce cost-dependent label noise, $\epsilon \sim \mathcal{N}(\bar{y}, \sigma^2)$, to each datapoint $\tilde{y}_i := y_i + \beta\tilde{\epsilon}/c_j$, where $\bar{y}$ is the mean target value and $\beta$ is the overall noise level, which we fix at $30\%$ throughout our experiments. For these experiments, we did not evaluate Data Shapley [23], LOO [13], and Influence [21] that had very long runtimes. In Table 2, we report additional mean test error across budgets 1–30 for both cost functions. We find that our DAVID method is more budget-efficient in choosing cost-effective noisy datapoints than other methods across datasets. We provide additional plots for heterogeneous costs in Appendix D.1.

**Comparing Runtime.** In Figure 5, we compare the optimization runtime of our data selection method on 1,000 datapoints while increasing the dimensionality of the data as well as when the dimensionality is fixed to 30, and the number of seller datapoints is increased to 100,000. Data Shapley [23] and LOO [13] took too long to run for large amounts of datapoints or high dimensional data and are not reported. In both experiments, our multi-step compares favorably to efficiency-optimized techniques such as KNN Shapley [29] while our single-step method had the fastest runtime. This demonstrates that our method can scale to marketplaces with millions of datapoints.

**Regularization Strength.** In Appendix D.2, we vary the amount of regularization applied on the MIMIC, DrugLib, and RSNA datasets. We find that applying a moderate amount of regularization between 0.2 and 0.6 can lead to improved performance. Even when the information matrix is set to identity, i.e., $\lambda = 1$, performance on the DrugLib datasets is still reasonable. Note that for all other experiments, we do not apply any regularization.

**Amount of Buyer Data.** In Appendix D.3, we vary how many buyer test datapoints are simultaneously optimized over on Gaussian-distributed, MIMIC, and RSNA datasets. While all buyer and seller data is sampled from the same distribution, the number of buyer datapoints still affects the optimization procedure. In general, we find that increasing the number of datapoints in the "test batch" increases

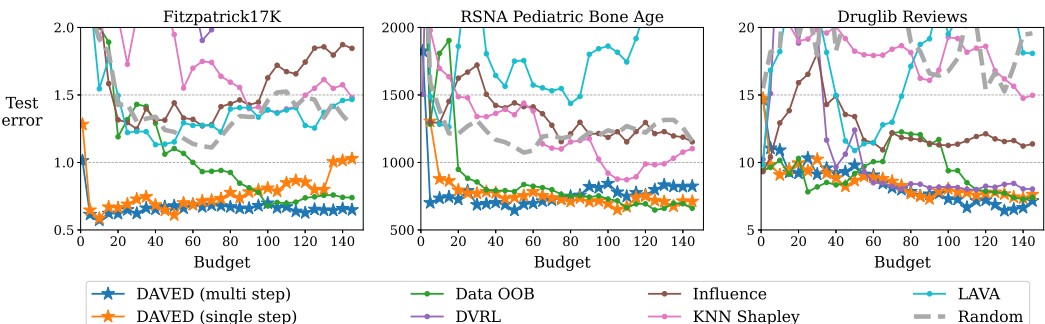

Figure 4: **Data Acquisition Performance on Real Medical Datasets.** DAVED demonstrates strong performance on real-world medical imaging and drug review datasets. (Left to right) Results on Fitzpatrick17K (skin lesions), RSNA Bone Age (X-rays), and DrugLib (drug reviews) — where high-dimensional raw data is embedded via CLIP (images) or GPT-2 (text). Each method selects training points under budget constraints to train a regression model on the embedded data. DAVED achieves lower test prediction error using fewer training points compared to validation-based approaches, demonstrating effectiveness on high-dimensional data.

Table 1: **Test Error of Data Valuation Methods.** We compared the test mean squared error on the buyer test point on a synthetic Gaussian-distributed data and four medical datasets: MIMIC, RSNA, Fitzpatrick17K, and DrugLib. The subheading denotes the number of seller training data available for that experiment, and "N/A" denotes that the method exceeded runtime constraints for the experiment. We optimize a separate random sample of training and validation data for each buyer and average over 100 buyers. Bolded values indicate the best-performing method and underlined values denote the second-best-performing method.

| Method | Gaussian | | MIMIC | | RSNA | Fitzpatrick | DrugLib |
|---|---|---|---|---|---|---|---|
| | 1K | 100K | 1K | 35K | 12K | 15K | 3.5K |
| Random baseline | 1.38 | 1.01 | 301.0 | 283.7 | 1309.1 | 1.49 | 21.4 |
| Data Shapley [23] | 0.87 | N/A | 294.9 | N/A | N/A | N/A | N/A |
| Leave One Out [13] | 1.31 | N/A | 1125.0 | N/A | N/A | N/A | N/A |
| Influence [21] | 1.47 | 0.97 | 189.4 | 876.4 | 1614.5 | 1.93 | 12.8 |
| DVRL [62] | 1.33 | 1.26 | 229.7 | 285.5 | 3528.8 | 3.00 | 12.6 |
| LAVA [32] | 1.47 | 1.10 | 190.9 | 417.3 | 1867.5 | 1.45 | 17.4 |
| KNN Shapley [29] | 1.55 | 1.18 | 175.7 | 229.6 | 1387.0 | 1.82 | 19.0 |
| Data OOB [39] | 1.24 | 0.98 | **169.7** | 215.6 | 1020.3 | 1.35 | 10.0 |
| DAVED (single step) | 0.58 | 0.27 | 277.4 | 659.9 | 900.2 | 0.73 | **9.0** |
| DAVED (multi-step) | **0.37** | **0.16** | 206.7 | **171.4** | **785.2** | **0.67** | 9.2 |

Table 2: **Test Error with Heterogeneous Costs.** Comparing data selection methods for two different cost functions, $\sqrt{c}$ and $c^2$. For each budget constraint, we select seller datapoints until the budget is exceeded and calculate test prediction error on the buyer data. We average over 100 buyers and report the mean test error across budgets from 1 to 30.

| | Gaussian | | MIMIC | | RSNA | | Fitzpatrick | | DrugLib | |
|---|---|---|---|---|---|---|---|---|---|---|
| COST FUNCTION | $\sqrt{c}$ | $c^2$ | $\sqrt{c}$ | $c^2$ | $\sqrt{c}$ | $c^2$ | $\sqrt{c}$ | $c^2$ | $\sqrt{c}$ | $c^2$ |
| Random baseline | 2.36 | 77.7 | 288 | 285 | 2254 | 2065 | 2.14 | 2.11 | 16.5 | 18.4 |
| DVRL | 1.67 | 2.1 | 214 | 215 | 24003 | 5588 | 1.46 | 8.90 | 22.5 | 20.7 |
| LAVA | 2.13 | 3.3 | 482 | 475 | 1667 | 1587 | 2.09 | 2.21 | 35.7 | 34.8 |
| KNN Shapley | 2.13 | 69.0 | 217 | 956 | 2754 | 2506 | 1.85 | 2.15 | 13.6 | 13.0 |
| Data OOB | 2.19 | 3.3 | 243 | 246 | 1695 | 1205 | 2.08 | 2.52 | 10.8 | 10.8 |
| DAVED (single) | 1.54 | 251.5 | 598 | 585 | 1734 | 1550 | **0.75** | **0.71** | **9.4** | **10.1** |
| DAVED (multi) | **0.04** | **0.2** | **169** | **168** | 1076 | **942** | 0.76 | 0.75 | 12.6 | 11.4 |

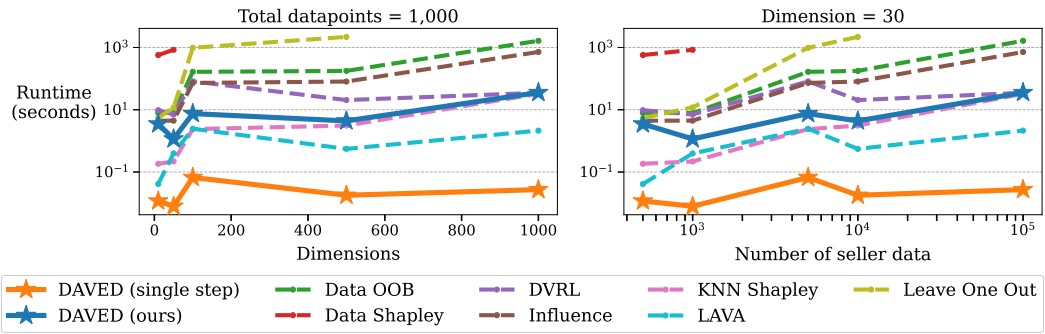

Figure 5: **Computational efficiency comparison.** DAVED has significantly lower computational overhead compared to model-based data valuation methods. (Left) Runtime scaling with data dimensionality (fixed 1,000 datapoints). (Right) Runtime scaling with the amount of seller data (fixed 30 dimensions). Our single-step variant is faster than even optimized methods like KNN Shapley, while the multi-step variant remains efficient while achieving better performance. Our optimization procedure only requires $O(d)$ communication per round, which makes it particularly suited for decentralized data market settings. For Data Shapley and Leave-One-Out, some experiments were omitted due to prohibitively long runtimes.

test errors. Therefore, we recommend keeping the number of test datapoints in the buyer's "query" between 1–8 for each data acquisition.

**Number of Steps.** In Appendix D.4, we vary the number of optimization steps in our method on the Gaussian-distributed and RSNA datasets. We find that more iterations generally improve prediction performance. Intuitively, one expects that selecting $T$ points requires at least $T$ steps of iterative optimization. We recommend setting the number of steps to be 2–5 times the desired budget for homogeneous costs.

**Convex versus Iterative Optimization** In Appendix D.6, we compare the iterative optimization procedure against a convex optimization solver [18]. We find that our iterative approach results in several orders of magnitude speedup while maintaining similar levels of test error.

**Finetuning versus Linear Probe.** In Figure 14, we evaluate fine-tuning versus linear probing for datapoints selected using DAVID and random selection. We find that using DAVID for fine-tuning performs similarly to linear probing results on DrugLib with BERT [17].

## 6 Discussion

While other validation-free methods exist [60, 5], our method uniquely combines test-adaptivity, theoretical grounding, and superior empirical performance. Moreover, a major advantage of our method is that it is amenable to federated optimization requiring $O(d)$ communication per round, making it well-suited for decentralized data marketplaces, unlike other methods that require seller data to be centralized in order to repeatedly train models to estimate data value. Additionally, our method does not require labeled data, whereas other data valuation methods assume that all datapoints come with corresponding ground-truth labels. As discussed in Section 2 and Section 3, the existing paradigm of valuing data with a validation set is suboptimal. Incidentally, the second-best performing method, Data OOB [39], is the only other method that does not use a validation set.

**Limitations.** However, our algorithm comes with some limitations that form exciting directions for future work. Our approach currently communicates every step. Instead, integrating local steps like in FedAvg [43] or Scaffold [35] would decrease communication costs. Further, integrating differential privacy techniques would provide formal privacy guarantees to the buyers and sellers [19]. While DAVED shows strong performance across datasets, its effectiveness depends on having a good feature extractor that captures relevant aspects of the data. We recommend using pre-trained foundation models (e.g., CLIP, GPT-2) as they can extract general-purpose features. Future work could explore adapting the feature extraction to specific domains or handling cases where key features are missing

## Acknowledgments

C.L. is funded by the National Science Foundation Graduate Research Fellowship Program. This material is based upon work supported by the National Science Foundation Graduate Research Fellowship Program under Grant No. 2141064. Any opinions, findings, and conclusions or recommendations expressed in this material are those of the author(s) and do not necessarily reflect the views of the National Science Foundation. S.P.K. and M.J. were funded by the European Union (ERC-2022-SYG-OCEAN-101071601). P.V. was partially supported by the ADIA Lab Fellowship.

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

# A  Proof of Theorem 1

**Theorem A.1.** *Let $\mathbf{w}^*$ denote the solution of Problem* (1) *and let $\hat{\mathbf{w}}$ denote the solution of Problem* (2). *Let the data $Z^{\mathrm{val}}$, $Z^{\mathrm{test}}$ are drawn i.i.d. from the distribution $\mathcal{D}_{X,Y}$ where $\mathcal{D}_X$ is supported on $B_R^d$ (zero-centered ball with radius $R$ in $\mathbb{R}^d$), and $Y = \theta^\top X + \eta$ where $\eta$ is independent zero-meaned noise with variance $\sigma^2$. Suppose $\mathcal{D}_X$ and the training data $X^{\mathrm{train}}$ is supported on $B_R^d$ (zero-centered ball with radius $R$ in $\mathbb{R}^d$) and $l$ is square loss, then there exist numerical constants, $c_1, c_2, c_3$, such that:*

1. *With probability at least $1 - \exp\left(-c_1 n_{\mathrm{val}}/R^2\right)$,*

$$\inf_{\hat{\theta}} \sup_{\mathcal{D}_{Y|X}, X^{\mathrm{train}}} \mathbb{E}_{X^{\mathrm{test}}} \left[\mathcal{L}(\hat{\mathbf{w}}) - \mathcal{L}(\mathbf{w}^*)\right] \geq \frac{c_2 \sigma^2 d}{n_{\mathrm{val}}}.$$

2. *If there exists $\kappa > 0$ such that $\lambda(\mathbb{E}_{x\sim\mathcal{D}_X}[x^{\otimes 4}]) \leq \kappa \cdot \lambda\left(\mathbb{E}_{x\sim\mathcal{D}_X}[x^{\otimes 2}]^{\otimes 2}\right)$ (here $\lambda$ denotes the largest eigenvalue and $\otimes$ denotes the outer product), then for any training algorithm used by the platform, with probability at least $0.99 - \exp\left(-c_1 n_{\mathrm{val}}/R^2\right) - \frac{c_3 \kappa}{\kappa + m}$, we have*

$$\sup_{\mathcal{D}_{Y|X}, X^{\mathrm{train}}} \mathcal{L}(\hat{\mathbf{w}}) - \mathcal{L}(\mathbf{w}^*) \geq \frac{c_2 \sigma^2 d}{n_{\mathrm{val}}}.$$

*Proof.* Let $f_\theta(x) = \theta^\top x$ and $\mathcal{E} = N(0, \sigma^2)$. Define the parameter space resulting from the training algorithm:

$$\Theta = \left\{ \hat{\theta}(\mathbf{w}) : \mathbf{w} \in \Delta([m]) \right\}.$$

When $n$ is sufficiently large, Lemma A.2 implies that there exists $\{\theta_1, \theta_2, \ldots, \theta_K\} \subset \Theta$ such that

$$\left\| \theta_i^\top X^{\mathrm{val}} \right\|_2 \lesssim \delta \sqrt{n_{\mathrm{val}}}, \ \forall i \in [K]$$
$$\left\| (\theta_i - \theta_j)^\top X^{\mathrm{val}} \right\|_2 \asymp \delta \sqrt{n_{\mathrm{val}}}, \ \forall i < j \in [K].$$

Let $\mathbb{P}_{\theta_i}$ denote the conditional distribution $\mathcal{D}_{y|x}$ of the target when the underlying model parameter is $\theta_i$, it then follows that

$$\mathrm{KL}(\mathbb{P}_{\theta_i} \| \mathbb{P}_{\theta_j}) \lesssim \frac{n\delta^2}{\sigma^2}.$$

Applying Lemma A.3, we obtain that

$$\inf_{\hat{\theta}} \sup_{\mathcal{D}_{Y|X}, X^{\mathrm{train}}} \mathbb{E}\left[ \frac{1}{n_{\mathrm{val}}} \left\| (\hat{\theta}(\hat{\mathbf{w}}) - \theta^*)^\top X^{\mathrm{val}} \right\|_2^2 \right] \gtrsim \frac{c_2 \sigma^2 d}{n_{\mathrm{val}}}. \tag{12}$$

Now define $\Sigma = \mathbb{E}_{x\sim\mathcal{D}_X}[xx^\top]$, by Lemma A.4, we have that with probability at least $1 - \exp(-\Omega(n_{\mathrm{val}}/R^2))$,

$$\frac{1}{2}\Sigma \preceq X^{\mathrm{val}}(X^{\mathrm{val}})^\top \preceq 2\Sigma. \tag{13}$$

Notice that

$$\mathbb{E}_{X^{\mathrm{test}}}\left[\mathcal{L}(\hat{\mathbf{w}}) - \mathcal{L}(\mathbf{w}^*)\right] = \left\| (\hat{\theta}(\hat{\mathbf{w}}) - \theta^*)^\top \right\|_\Sigma^2.$$

Therefore, under the event of Eq. (12), Eq. (13) implies that

$$\inf_{\hat{\theta}} \sup_{\mathcal{D}_{Y|X}, X^{\mathrm{train}}} \mathbb{E}_{X^{\mathrm{test}}}\left[\mathcal{L}(\hat{\mathbf{w}}) - \mathcal{L}(\mathbf{w}^*)\right] \geq \inf_{\hat{\theta}} \sup_{\mathcal{D}_{Y|X}, X^{\mathrm{train}}} \frac{1}{2} \mathbb{E}\left[ \frac{1}{n_{\mathrm{val}}} \left\| (\hat{\theta}(\hat{w}) - \theta^*)^\top X^{\mathrm{val}} \right\|_2^2 \right]$$
$$\gtrsim \frac{c_2 \sigma^2 d}{n_{\mathrm{val}}}$$

This establishes the first inequality.

For the second inequality, we have that under the condition $\lambda(\mathbb{E}_{x\sim\mathcal{D}_X}[x^{\otimes 4}]) \leq \kappa \cdot \lambda\left(\mathbb{E}_{x\sim\mathcal{D}_X}[x^{\otimes 2}]^{\otimes 2}\right)$, the following holds

$$
\begin{aligned}
\mathrm{Var}_{x\sim D_X}\left(\left\langle\hat{\theta}(\hat{\mathbf{w}})-\theta^*,x\right\rangle^2\right) &= \mathbb{E}\left[\left\langle\hat{\theta}(\hat{\mathbf{w}})-\theta^*,x\right\rangle^4\right] - \mathbb{E}\left[\left\langle\hat{\theta}(\hat{\mathbf{w}})-\theta^*,x\right\rangle^2\right]^2 \\
&= \mathbb{E}\left[\left\langle(\hat{\theta}(\hat{\mathbf{w}})-\theta^*)^{\otimes 4},x^{\otimes 4}\right\rangle\right] - \mathbb{E}\left[\left\langle(\hat{\theta}(\hat{\mathbf{w}})-\theta^*)^{\otimes 2},x^{\otimes 2}\right\rangle^2\right] \\
&= \left\langle(\hat{\theta}(\hat{\mathbf{w}})-\theta^*)^{\otimes 4},\mathbb{E}\left[x^{\otimes 4}\right]-\mathbb{E}\left[x^{\otimes 2}\right]^{\otimes 2}\right\rangle \\
&\leq (\kappa-1)\cdot\mathbb{E}\left[\left\langle\hat{\theta}(\hat{\mathbf{w}})-\theta^*,x\right\rangle^2\right]^2.
\end{aligned}
$$

By Lemma A.5, for any $\theta$ and $\theta^*$, we have that with probability at least $0.99 - O\left(\frac{\kappa}{\kappa+m}\right)$,

$$
\begin{aligned}
\mathcal{L}(\hat{\mathbf{w}}) - \mathcal{L}(\mathbf{w}^*) &= \frac{1}{m}\left\|(\theta-\theta^*)^\top X^{\text{test}}\right\|_2^2 \\
&= \frac{1}{m}\sum_{i=1}^m\left\langle\theta-\theta^*,x_i^{\text{test}}\right\rangle^2 \\
&\geq 0.0001\cdot\left\|(\hat{\theta}(\hat{\mathbf{w}})-\theta^*)^\top\right\|_\Sigma^2 \\
&\gtrsim \frac{c_2\sigma^2 d}{n_{\text{val}}}.
\end{aligned}
$$

Combining this and the first inequality by union bound, we establish the second inequality. $\qquad\square$

## A.1   Supporting Lemma

**Lemma A.2** (Metric entropy, Wainwright [56]). *Let $\|\cdot\|$ denote the Euclidean norm on $\mathbb{R}^d$ and let $\mathbb{B}$ be the unit balls (i.e., $\mathbb{B} = \{\theta\in\mathbb{R}^d|\|\theta\|\leq 1\}$). Then the $\delta$-covering number of $\mathbb{B}$ in the $\|\cdot\|$-norm obeys the bounds*

$$
d\log\left(\frac{1}{\delta}\right) \leq \log N(\delta;\mathbb{B},\|\cdot\|) \leq d\log\left(1+\frac{2}{\delta}\right).
$$

**Lemma A.3** (Fano's inequality, Cover [14]). *When $\theta$ is uniformly distributed over the index set $[M]$, then for any estimator $\hat{\theta}$ such that $\theta\to Z\to\hat{\theta}$*

$$
\mathbb{P}[\hat{\theta}(Z)\neq\theta] \geq 1 - \frac{I(Z;\theta)+\log 2}{\log M}.
$$

**Lemma A.4** (Matrix-Chernoff bound, Tropp [55]). *Consider an independent sequence $\{X_i\}_{i=1}^k$ of random, self-adjoint matrices in $M_n$ satisfying $X_i\geq 0$ and $\lambda_{max}(X_i)\leq R$ almost surely, for each $i\in\{1,\dots,k\}$. Define*

$$
\mu_{min} := \lambda_{min}\left(\sum_{i=1}^k\mathbb{E}X_i\right),
$$

$$
\mu_{max} := \lambda_{max}\left(\sum_{i=1}^k\mathbb{E}X_i\right).
$$

*Then,*

$$
\mathbb{P}\left\{\lambda_{max}\left(\sum_{i=1}^k X_i\right)\geq(1+\delta)\mu_{max}\right\} < n\left(\frac{\mathrm{e}^\delta}{(1+\delta)^{1+\delta}}\right)^{\frac{\mu_{max}}{R}} \quad \text{for } \delta\geq 0;
$$

$$
\mathbb{P}\left\{\lambda_{min}\left(\sum_{i=1}^k X_i\right)\leq(1-\delta)\mu_{min}\right\} < n\left(\frac{\mathrm{e}^{-\delta}}{(1-\delta)^{1-\delta}}\right)^{\frac{\mu_{min}}{R}} \quad \text{for } \delta\in[0,1].
$$

**Lemma A.5** (Paley–Zygmund inequality, Paley and Zygmund [45]). *If $Z$ is a random variable with finite variance and $Z \geq 0$ almost surely, then*

$$\mathbb{P}(Z > \theta \, \mathbb{E}[Z]) \geq (1 - \theta)^2 \frac{\mathbb{E}[Z]^2}{\mathbb{E}[Z^2]}, \ \forall \theta \in (0, 1).$$

# B Convergence Rate of Frank-Wolfe (Proof of Theorem 2

**Setup.** We mostly follow the proof technique in Bach et al. [8] and Jaggi [28]. Recall the optimization problem for the optimal design loss Eq. (4):

$$\min_{\mathbf{w} \in \mathcal{D}} \mathcal{L}(\mathbf{w}).$$

Here, $\mathcal{D}$ is the scaled simplex defined by the constraints $\mathbf{w} \in \mathbb{R}_{\geq 0}^n$ and $\sum_{j=1}^n c_j w_j \leq B$. The Frank-Wolfe update on this function is then: For $t = 1, 2, \ldots$, repeatedly perform the following steps

- Compute $\mathbf{s}_t = \arg\max_{\mathbf{u} \in \mathcal{D}} \langle \nabla \mathcal{L}(\mathbf{w}_t), \mathbf{w}_t - \mathbf{u} \rangle$.
- update $\mathbf{w}_{t+1} = (1 - \alpha_t)\mathbf{w}_t + \alpha_t \mathbf{s}_t$

Note that this update procedure is identical to the updates in Eq. 5. We can then define the duality gap as

$$g(\mathbf{w}) = \sup_{\mathbf{s} \in \mathcal{D}} \langle \mathbf{w} - \mathbf{s}, \nabla \mathcal{L}(\mathbf{w}) \rangle.$$

We also define the curvature constant

$$C_l = \sup_{\substack{\mathbf{s}, \mathbf{w} \in \mathcal{D} \\ \gamma \in (0,1) \\ \mathbf{u} = (1-\gamma)\mathbf{w} + \gamma\mathbf{s}}} \frac{2}{\gamma^2} \left( \mathcal{L}(\mathbf{u}) - \mathcal{L}(\mathbf{w}) - \langle \mathbf{u} - \mathbf{w}, \nabla \mathcal{L}(\mathbf{w}) \rangle \right).$$

We assume that the curvature constant is finite, i.e., $C_l < \infty$. This is true for $\mathcal{L}(\mathbf{w})$ as long as both the algorithm and the true optimum are bounded away from the boundary of $\mathcal{D}$ — see detailed discussions on this in Ahipaşaoğlu and Todd [2]. A better analysis might be able to avoid this assumption, e.g., Zhao and Freund [65] use certain homogeneity properties of $\mathcal{L}(\mathbf{w})$ to derive better assumption-free convergence rates for the FW method on D-optimal experiment design. We leave the question of adapting these results to our setting (V-optimal experiment design) for a challenging future work.

**Lemma B.1** (Lemma 5, [28]). *For any $\alpha \in (0, 1)$,*

$$\mathcal{L}(\mathbf{w}_{t+1}) \leq \mathcal{L}(\mathbf{w}_t) - \alpha_t g(\mathbf{w}_t) + \frac{\alpha_t^2}{2} C_l.$$

**Theorem B.2.** *In the Frank-Wolfe algorithm, our update algorithm in Eq. 5 uses $\alpha_t = \frac{1}{t+1}$. For this, we have*

$$\mathcal{L}(\mathbf{w}_t) \leq \min_{\mathbf{w} \in \mathcal{D}} \mathcal{L}(\mathbf{w}) + \frac{C_l(1 + \log t)}{2t}.$$

*Proof.* Define $h(\mathbf{w}) = \mathcal{L}(\mathbf{w}) - \min_{\mathbf{w} \in \mathcal{D}} \mathcal{L}(\mathbf{w})$. Using Lemma B.1,

$$h(\mathbf{w}_t) \leq h(\mathbf{w}_{t-1}) - \alpha_{t-1} g(\mathbf{w}_{t-1}) + \frac{\alpha_{t-1}^2}{2} C_l$$

$$\leq h(\mathbf{w}_{t-1}) - \alpha_{t-1} h(\mathbf{w}_{t-1}) + \frac{\alpha_{t-1}^2}{2} C_l$$

$$= (1 - \alpha_{t-1}) h(\mathbf{w}_{t-1}) + \frac{\alpha_{t-1}^2}{2} C_l$$

Here we used the convexity of $\mathcal{L}$ as follows:

$$g(\mathbf{w}_{t-1}) = \sup_{\mathbf{s} \in \mathcal{D}} \langle \mathbf{w}_{t-1} - \mathbf{s}, \nabla \mathcal{L}(\mathbf{w}_{t-1}) \rangle$$

$$\geq \langle \mathbf{w}_{t-1} - \mathbf{w}^*, \nabla \mathcal{L}(\mathbf{w}_{t-1}) \rangle$$

$$\geq \mathcal{L}(\mathbf{w}_{t-1}) - \min_{\mathbf{w} \in \mathcal{D}} \mathcal{L}(\mathbf{w}).$$

Continuing our derivation, recall we have

$$h(\mathbf{w}_t) \leq (1 - \alpha_{t-1})h(\mathbf{w}_{t-1}) + \frac{\alpha_{t-1}^2}{2}C_l$$
$$= \frac{t-1}{t}h(\mathbf{w}_{t-1}) + \frac{1}{2t^2}C_l \quad \leq \quad \frac{t-2}{t}h(\mathbf{w}_{t-1}).$$

Thus we have

$$t \cdot h(\mathbf{w}_t) \leq (t-1)h(\mathbf{w}_{t-1}) + \frac{C_l}{2t}.$$

Unrolling this recursion, we get

$$t \cdot h(\mathbf{w}_t) \leq \sum_{k=1}^{t-1} \sum \frac{C_l}{2k} \leq \frac{C_l(1 + \log t)}{2}.$$

This yields the theorem claim.  □

Finally, to finish the proof of Theorem 2, note that adding additional constraints only increases the loss, i.e.,

$$\min_{\mathbf{w} \in \mathcal{D}} \mathcal{L}(\mathbf{w}) \leq \min_{\mathbf{w} \in \mathcal{D} \text{ and } \mathbf{w} \in \{0,1\}^n} \mathcal{L}(\mathbf{w}).$$

Hence, we showed a $O(\log t/t)$ approximation to the otherwise intractable combinatorial problem. With additional assumptions on the structure of the loss function, one can even show improved quadratic or even exponential approximations [15, 8]. Finally, we note that Frank-Wolfe is a well-known method to efficiently approximate the optimal experiment design objective [59, 20, 2] and this also motivates using this approach in practice [2, 6].

## C   Experimental Setup

For each buyer test point, we optimize each selection algorithm over the 1,000 seller datapoints and select the highest value data based on the validation set of 100 datapoints (our method and Data OOB do not use the validation set). For each test point, we train a linear regression model on the selected seller points and report test mean squared error (MSE) on the buyer's data and average test error over 100 buyers.

For reproducibility, our full implementation is available at: `https://github.com/clu5/data-acquisition-via-experimental-design`.

### C.1   Implementation Details

We conduct all experiments on an Intel Xeon E5-2620 CPU with 40 cores and a Nvidia GTX 1080 Ti GPU. For implementation of baseline data valuation methods, we use the OpenDataVal package [30] version 1.2.1. We use the default hyperparameter settings for all methods except for Data Shapley (changed 100 Monte-Carlo epochs with 10 models per iteration), Influence Subsample (from 1000 to 500 models), and Data OOB (from $1,000$ to $500$ models) to reduce computational runtime.

In our experiments, we use the following setting of hyperparameters for DAVED:

- 500 iterations for multi-step variant, 1 iteration for single-step variance
- Line search for step size $\alpha \in (0, 0.9)$
- Regularization $\lambda = 0$ (unless otherwise specified)
- No early stopping

For experiments with heterogeneous costs, we uniformly sample costs $c \in \{1, 2, 3, 4, 5\}$ for each seller datapoint and apply either $h(c) = \sqrt{c}$ or $h(c) = c^2$ as the cost function. Noisy labels are generated by adding Gaussian noise scaled inversely proportional to the cost, with overall noise level set to 30%.

## C.2 Dataset Details and Processing

For Fitzpatrick17K and RSNA Bone Age datasets, each image was embedded through a CLIP ViT-B/32 model [49], while for the DrugLib dataset, each text review was embedded through GPT-2 model [48] with a max context length of 4096.

For the Gaussian dataset, we generate a regression dataset according to the following Python code:

```python
import numpy as np

def get_gaussian_data(num_samples=100, dim=10, noise=0.1, costs=None):
    X = np.random.normal(size=(num_samples, dim))
    X /= np.linalg.norm(X, axis=1, keepdims=True)
    if costs is not None:
        X *= costs
    coef = np.random.exponential(scale=1, size=dim)
    coef *= np.sign(np.random.uniform(low=-1, high=1, size=dim))
    y = X @ coef + noise * np.random.randn(num_samples)
    return dict(X=X, y=y, coef=coef, noise=noise, dim=dim, costs=costs)
```

The RSNA Pediatric Bone Age Challenge (2017) dataset [25] may be downloaded here https://www.rsna.org/rsnai/ai-image-challenge/rsna-pediatric-bone-age-challenge-2017. We use the training set for our experiments, resulting in 12,611 images in total. Using the following function, each image was embedded through a pre-trained CLIP ViT-B/32 model.

```python
import clip
import torch
from PIL import Image

def embed_images(img_paths, device="cuda"):
    model, preprocess = clip.load("ViT-B/32", device=device)
    inference_func = model.encode_image
    embeddings = []
    with torch.inference_mode():
        for img_path in tqdm(img_paths):
            img = Image.open(img_path)
            embedding = inference_func(preprocess(img)[None].to(device))
            embeddings.append(embedding.cpu())
    return torch.cat(embeddings)
```

The Fitzpatrick17K [24] can be downloaded from here https://github.com/mattgroh/fitzpatrick17k. Missing or corrupted images were excluded, resulting in 16,536 total images. Some images were annotated with two separate annotation platforms. We averaged the two skin type ratings for these images, resulting in 12 possible labels (0–6 in 0.5 increments). Each image was embedded in a fashion similar to the RSNA Bone Age dataset.

The MIMIC dataset [31] can be accessed here https://physionet.org/content/mimiciii/1.4/. The task is to predict the length of stay (LOS) in the number of days a patient stays in the Intensive Care Unit. The dataset contains 51,036 rows with both real-valued and one-hot-encoded attributes with the following names:

> "LOS" , "blood" , "circulatory" , "congenital" , "digestive" , "endocrine" , "genitourinary" , "infectious" , "injury" , "mental" , "misc" , "muscular" , "neoplasms" , "nervous" , "pregnancy" , "prenatal" , "respiratory" , "skin" , "GENDER" , "ICU" , "NICU" , "ADM_ELECTIVE" , "ADM_EMERGENCY" , "ADM_NEWBORN" , "ADM_URGENT" , "INS_Government" , "INS_Medicaid" , "INS_Medicare" , "INS_Private" , "INS_Self Pay" , "REL_NOT SPECIFIED" , "REL_RELIGIOUS" , "REL_UNOBTAINABLE" , "ETH_ASIAN" , "ETH_BLACK/AFRICAN AMERICAN" , "ETH_HISPANIC/LATINO" , "ETH_OTHER/UNKNOWN" , "ETH_WHITE" , "AGE_middle_adult" , "AGE_newborn" , "AGE_senior" , "AGE_young_adult" , "MAR_DIVORCED" , "MAR_LIFE PARTNER" , "MAR_MARRIED" , "MAR_SEPARATED" , "MAR_SINGLE" , "MAR_UNKNOWN (DEFAULT)" , "MAR_WIDOWED"

Each attribute was min-max scaled to lie in the range [0, 1].

The DrugLib dataset [34] can be downloaded here `https://archive.ics.uci.edu/dataset/461/drug+review+dataset+druglib+com`. The task is to predict overall patient satisfaction with a drug's side effects and effectiveness on a ten-point scale. We format each review using the following prompt template to feed GPT-2:

```
Benefits: $BENEFITS_REVIEW
Side effects: $SIDE_EFFECTS_REVIEW
Comments: $COMMENTS_REVIEW
```

where `$BENEFITS_REVIEW` is the corresponding portion of the drug review.

```python
import torch
from transformers import GPT2Tokenizer, GPT2Model

def embed_text(text_inputs :list[str], device="cuda"):
    tokenizer = GPT2Tokenizer.from_pretrained(model_name)
    model = GPT2Model.from_pretrained(model_name).to(device)
    embeddings = []
    for x in tqdm(text_inputs):
        inputs = tokenizer(x, return_tensors="pt", truncation=True).to(
    device)
        with torch.no_grad():
            outputs = model(**inputs)
        embeddings.append(
            outputs.last_hidden_state.mean(dim=1).cpu()
        )
    return torch.cat(embeddings)
```

## C.3 Evaluation Protocol

For all experiments (unless otherwise specified), we use the following evaluation settings:

- Training split: Variable number of seller points (1K–100K)
- Validation split: 100 points for baseline data valuation methods
- Test split: 100 buyer points evaluated independently
- Metrics: Mean squared error averaged over all buyer points
- Budget ranges: 1–30 for cost experiments, 1–150 otherwise Number of random seeds: All results averaged over 3 seeds

For a fair comparison, we use the same train/validation/test splits across all methods for each random seed. The validation set is only used by data valuation baseline methods that require it — our method DAVED operates directly on the test queries without requiring validation data.

# D   Additional Experiments

## D.1   Budget Results with Heterogeneous Costs

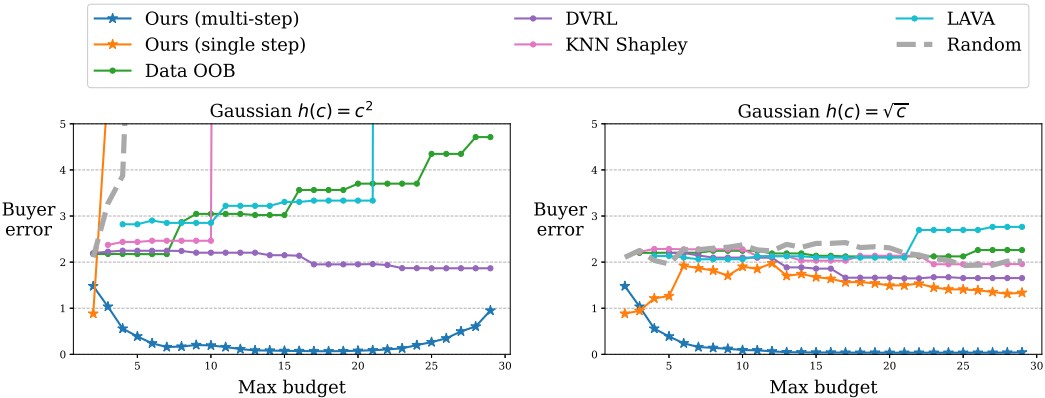

Figure 6: **DAVED maintains performance advantage with heterogeneous costs on synthetic data.** We evaluate data selection methods on 10K synthetic Gaussian datapoints with costs randomly sampled from $\{1, 2, 3, 4, 5\}$ and transformed using two cost functions: $\sqrt{c}$ (left) and $c^2$ (right). Each datapoint's label noise is scaled inversely with cost ($\varepsilon/c_j$) to simulate quality differences. DAVED achieves lower test MSE across budgets 1-30 while respecting cost constraints, demonstrating robust performance even with varying data quality and costs. Results are averaged over 100 random buyer test points.

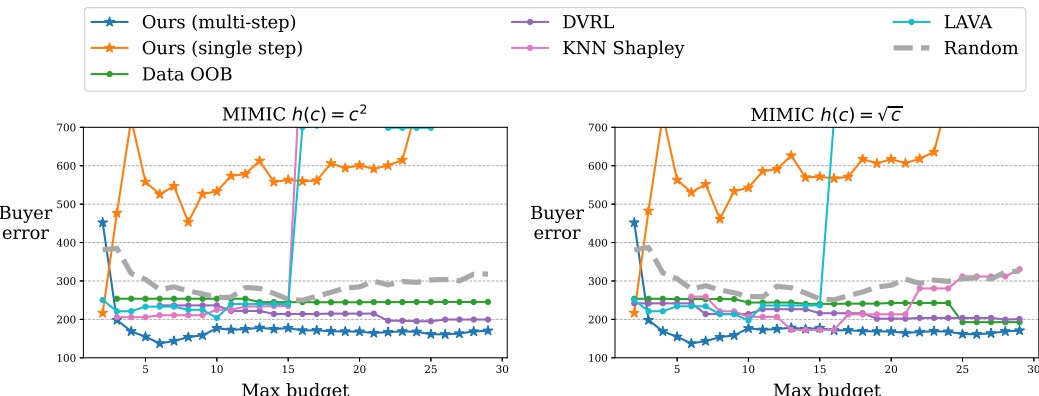

Figure 7: **Comparing performance on MIMIC healthcare data with heterogeneous costs.** Experiments on MIMIC-III dataset (35K patients with $48$ normalized clinical features) compare performance under two cost functions: $\sqrt{c}$ (left) and $c^2$ (right). Costs are uniformly sampled from $1, 2, 3, 4, 5$ per datapoint. Label noise $\varepsilon \sim \mathcal{N}(0, \sigma^2)$ is scaled by $\bar{y}/h(c_j)$ where $\bar{y}$ is the mean label value and $h(c)$ is the cost function, modeling how higher-cost data has lower noise. DAVED achieves lower prediction error for length-of-stay prediction across budgets 1-30. Results averaged over 100 random test patients.

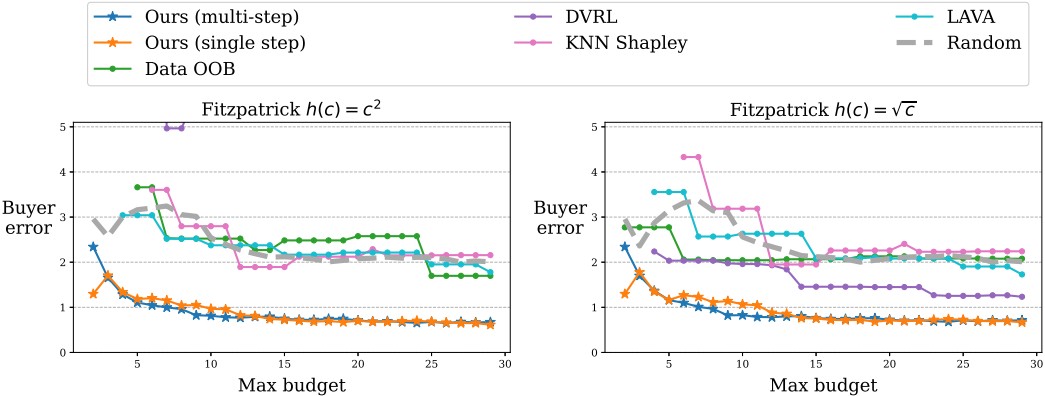

Figure 8: **Comparing performance on Fitzpatrick skin lesion dataset with varying costs.** Evaluation on Fitzpatrick17K dataset (15K images embedded through CLIP) under two cost functions: $\sqrt{c}$ (left) and $c^2$ (right). Costs sampled from $1, 2, 3, 4, 5$ with noise $\varepsilon \sim \mathcal{N}(0, \sigma^2)$ scaled by $\bar{y}/h(c_j)$, where $\bar{y}$ is the mean Fitzpatrick score and $h(c)$ is the cost function. This scaling ensures higher-cost images have proportionally less label noise. DAVED achieves consistent performance while respecting heterogeneous cost constraints. Results averaged over 100 random test images.

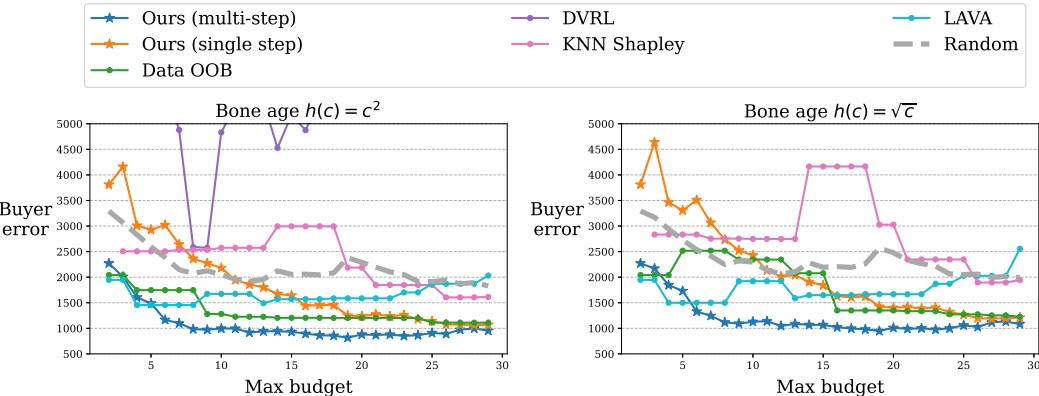

Figure 9: **Comparing performance on RSNA bone age prediction with cost variations.** Analysis on RSNA Bone Age dataset (12K X-ray images embedded via CLIP) using cost functions $\sqrt{c}$ (left) and $c^2$ (right). With costs sampled from $1, 2, 3, 4, 5$ and label noise $\varepsilon \sim \mathcal{N}(0, \sigma^2)$ scaled by $\bar{y}/h(c_j)$, where $\bar{y}$ is the mean age and $h(c)$ is the cost function. This models how higher-cost X-rays have more accurate age labels. DAVED maintains superior prediction accuracy compared to other data valuation baselines. Results show mean error over 100 random test X-rays.

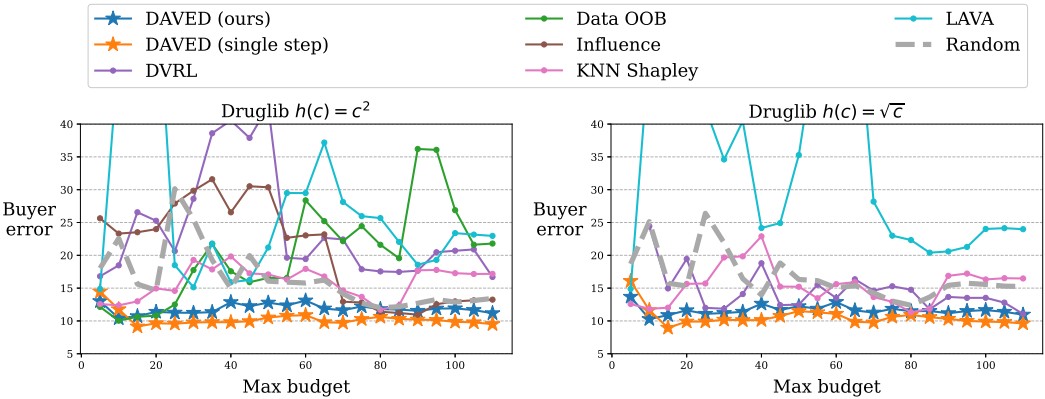

Figure 10: **Comparing performance on drug review text data with heterogeneous costs.** Evaluation on DrugLib reviews (3.5K reviews embedded using GPT-2) under cost functions $\sqrt{c}$ and $c^2$. Costs sampled from $1, 2, 3, 4, 5$ with noise $\varepsilon \sim \mathcal{N}(0, \sigma^2)$ scaled by $\bar{y}/h(c_j)$, where $\bar{y}$ is the mean rating and $h(c)$ is the cost function, reflecting how higher-cost reviews tend to have more reliable ratings. DAVED achieves robust rating prediction performance despite varying data quality. Results averaged over 100 random test reviews.

## D.2 Regularization

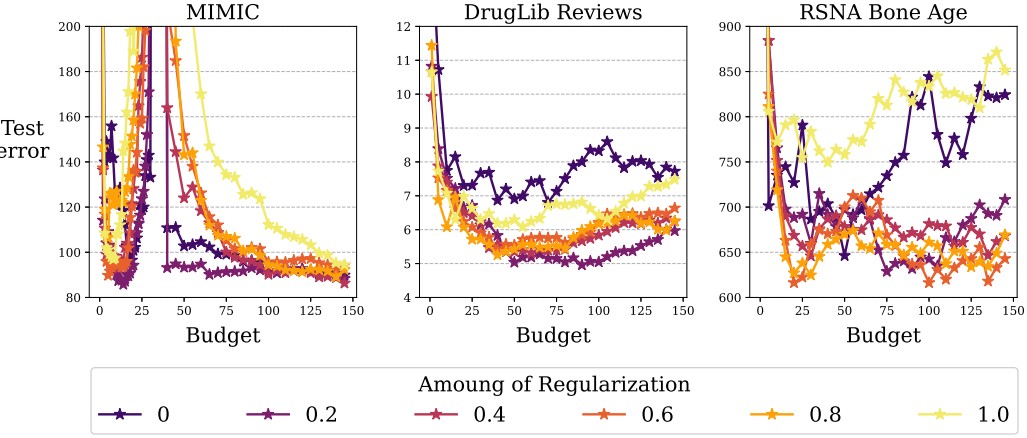

Figure 11: **Regularization strength significantly impacts DAVED performance across domains.** We evaluate prediction error versus budget as $\lambda$ varies from 0 (no regularization) to 1 (identity information matrix) on: MIMIC dataset (length-of-stay prediction), DrugLib reviews (rating prediction with GPT-2 embeddings), and RSNA Bone Age (age prediction from X-rays using CLIP embeddings). Moderate regularization ($\lambda$ between 0.2–0.6) improves stability and performance by better conditioning the information matrix, though DAVED remains effective even without regularization. Results averaged over 100 random test points per budget level.

## D.3    Amount of Buyer Data

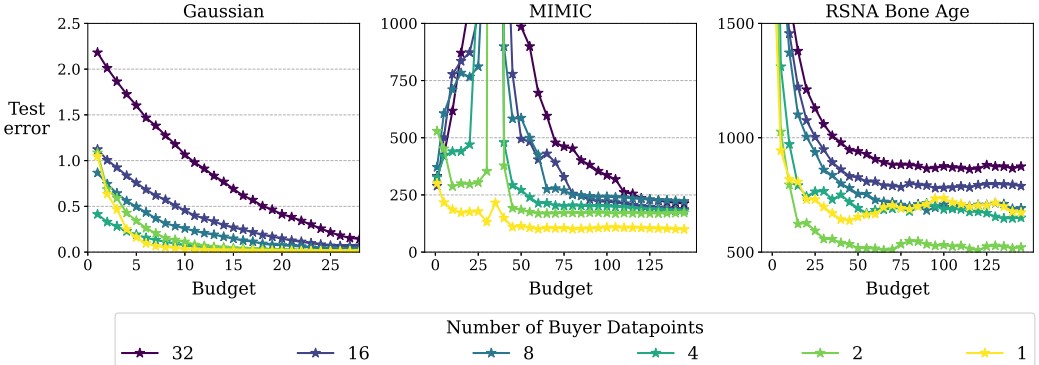

Figure 12: **Buyer test set size affects prediction performance of DAVED.** We vary the number of simultaneous test points (1 to 32) being optimized over for: Gaussian synthetic data (30 dimensions), MIMIC clinical data (48 dimensions), and RSNA Bone Age X-rays (512 dimensions). While all buyer and seller data follow the same distribution within each dataset, increasing test batch size leads to higher prediction errors as the optimization problem becomes more challenging. Results suggest keeping buyer queries to 1-8 test points for optimal performance. Each curve shows mean error over 100 random trials with different test sets.

## D.4    Number of Iteration Steps

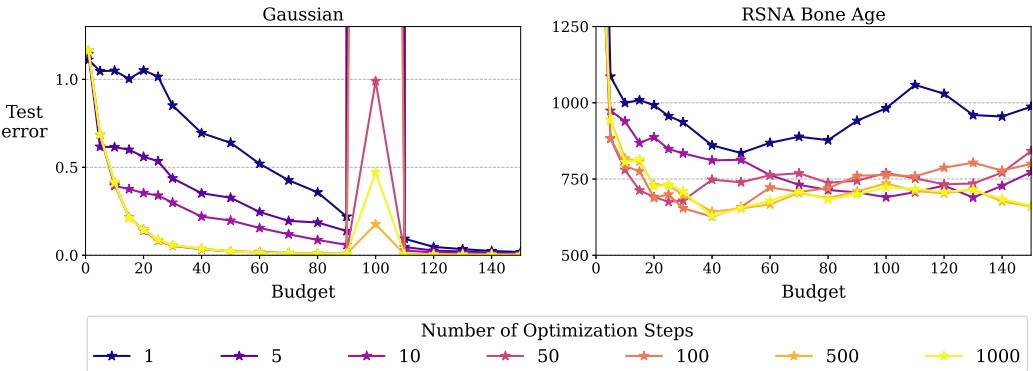

Figure 13: **Number of optimization steps impacts DAVED performance.** We evaluate prediction error versus budget while varying optimization steps from 1 to $1,000$ on Gaussian data (30 dimensions) and RSNA Bone Age dataset (embedded through CLIP). More iterations generally improve performance as the algorithm better approximates the optimal selection weights. For homogeneous costs ($c_j = 1$), we recommend using 2-5 times more optimization steps than the desired budget to ensure convergence. Results show mean error over 100 random test points per configuration.

## D.5 Linear Feature Space Approximation

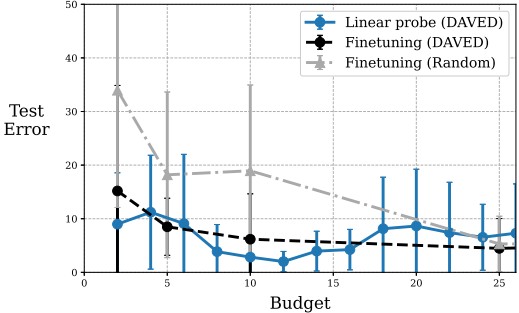

Figure 14: **Linear models in feature space approximate full model fine-tuning.** We compare BERT performance on DrugLib reviews using either full fine-tuning (which updates all model parameters) or linear probing (which only trains a linear layer on frozen embeddings) on data selected by DAVED versus random selection. The similar performance patterns between these approaches empirically validates our use of kernelized linear regression in the feature space as a proxy for the full training dynamics. This aligns with recent theoretical work showing that fine-tuning of pre-trained models is well-approximated by linear models in the empirical Neural Tangent Kernel (eNTK) regime. Since our method relies on this linear approximation for efficient data selection, these results support our choice of feature extractor $\phi$ and linear modeling approach. Results show test error averaged over 100 random trials using different test reviews.

## D.6 Iterative versus Convex optimization

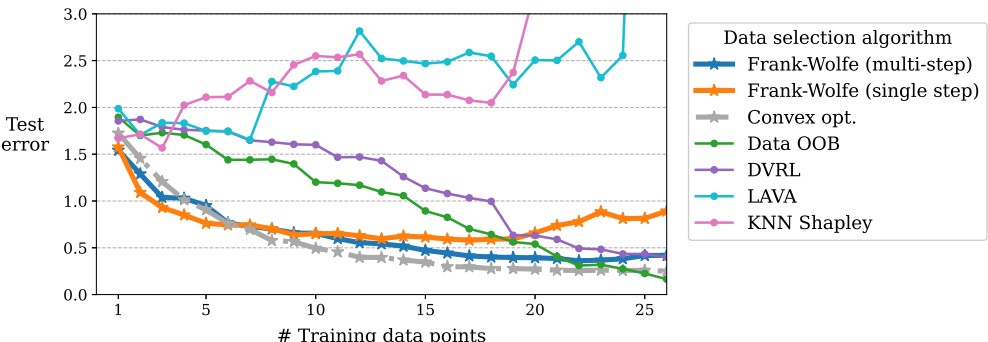

Figure 15: **DAVED's iterative optimization matches convex solver accuracy.** We compare the prediction error of Frank-Wolfe optimization with a convex optimization solver on $1,000$ datapoints sampled from 30-dimensional Gaussian distribution. Both single-step and multi-step variants of our iterative approach achieve comparable accuracy to the optimal convex solution while being significantly faster to optimize. Results averaged over 100 random trials with homogeneous costs ($c_j = 1$).

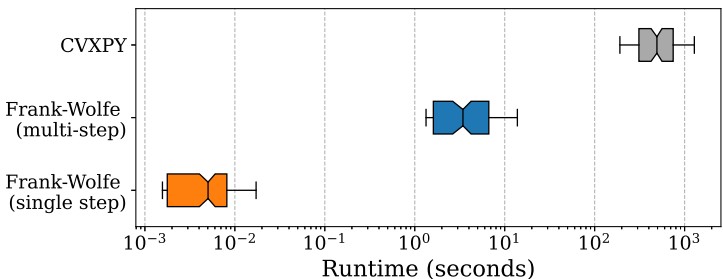

Figure 16: **DAVED provides orders of magnitude runtime speedup over convex optimization.** Runtime comparison between Frank-Wolfe iterative approach and convex solver for the data selection problem on 1000 datapoints from 30-dimensional Gaussian distribution. Our iterative method achieves several orders of magnitude speedup while maintaining similar levels of prediction error (see Figure 15). The single-step variant provides additional acceleration with minimal performance loss

## E   Broader Impacts

We believe that AI developers face important ethical questions when acquiring data for AI development, as highlighted by recent class-action lawsuits against AI companies regarding data consent and compensation. Our work presents a scalable and decentralized approach to data acquisition that can help address these concerns. By enabling targeted selection of the most valuable datapoints, our method reduces both costs and potential privacy risks compared to broad, indiscriminate data access [44]. The decentralized nature of our approach enhances transparency and gives individual data owners more control over how their data is shared and used, while avoiding the privacy and security risks of centralizing data with brokers. However, while our method enables more efficient and privacy-preserving data transactions that could democratize access to high-quality training data, especially in domains like healthcare, several societal and technical challenges remain. These include ensuring fair compensation for data owners, preventing misuse of acquired data, and developing appropriate governance frameworks for decentralized data markets. Addressing these challenges will be crucial for the responsible development of data marketplaces that can sustainably increase the supply of training data while protecting individual privacy and data rights.

