# OpenReview forum: "Data Acquisition via Experimental Design for Data Markets"
_NeurIPS.cc/2024/Conference — NeurIPS 2024 poster_

### Official Review · Reviewer_yTne · 2024-06-21

**Soundness:** 3
**Presentation:** 4
**Contribution:** 3
**Rating:** 7
**Confidence:** 5

**Summary:**

The paper focuses on the problem of data acquisition in decentralized data marketplaces. The paper claims that acquiring training data using validation-based data valuation methods can be overfitting. Thus, the paper proposes DAVED, which directly selects training data by optimizing test loss on a given test set rather than relying on the validation set. DAVED approximates the test loss trained on a subset of the training data using a linear experimental design framework and employs gradient-based optimization to compute the sampling probability of each data point, considering budget constraints and decentralized settings. Experiments on synthetic and real-world datasets are conducted to validate the advantages of DAVED.

**Strengths:**

(S1) The paper studies an interesting and important problem.

(S2) The paper proposes a novel approach for data acquisition in decentralized data marketplaces.

(S3) Experiments show the effectiveness of the proposed approach.

**Weaknesses:**

(W1) Lack of comparison with data acquisition approaches.

(W2) Lack of experimental evaluation on the test data distribution.

(W3) The motivation could be better clarified.

**Questions:**

(D1) The paper claims that existing data valuation methods are misaligned with the data acquisition problem and introduces DAVED to select data points based on the test set rather than the validation set. In the experiments, the paper compares DAVED with a series of data valuation methods including Influence, Data Shapley, and Data OOB, showing the advantages of DAVED. However, the paper does not discuss or compare its performance with methods that focus on data acquisition problems with or without a validation set. In addition, given that DAVED is designed for decentralized data marketplaces, the paper does not discuss or verify how it compares with existing methods in federated learning, such as [62].


(D2) The paper does not conduct experiments to analyze the impact of different test data distributions and the training data distribution. The paper should include experiments to measure the performance of DAVED under the scenario where there is a distribution shift between the training and test sets.

(D3) The paper uses an example to illustrate their setting: "the buyer may be a patient who cares about diagnosing their own chest X-ray and is willing to pay a fixed budget to access other X-ray images to train a model to have low error on their 'test set' (see Figure 1)." However, this example may be impractical as it is unlikely that a patient would train a diagnostic model themselves rather than relying on a hospital or healthcare institution. If the data acquisition is indeed for a hospital, the test data for the hospital cannot be fixed because patients are continually coming in. From my understanding, Algorithm 1 is designed for a fixed size of test data. This creates a gap between the paper's setting and real-world scenarios. The authors should either clarify this gap or explain if I have misunderstood the motivation or algorithm.

---

> ### Author Rebuttal · Authors · 2024-08-05
>
> We greatly thank the reviewer for their close reading of our work, their detailed feedback, and their enthusiastic support. We address the comments raised below.
>
> > [...] compare its performance with methods that focus on data acquisition problems
>
> First, we want to emphasize that our method uses an *unlabelled* test dataset whereas the previous methods need a labeled validation dataset. In this paper, we tried to show the limitations of such data valuation approaches but we agree that comparing against other federated client selection methods will be important for more realistic decentralized market scenarios. As we note in our limitations section, adapting our methods to a realistic communication-constrained federated learning setting is an important future direction we hope to explore.
>
> > [...] distribution shift between the training and test sets
>
> Note that we make no distributional assumptions on our data. In fact, this is one of the biggest strengths of our method. Both our derivation in Step 2 (line 149) as well as our guarantee in Theorem 2 holds for an arbitrary test dataset - it may be completely different from the training dataset. The only assumption we make is on that the *conditional distribution* $Y | X$ is the same  i.e. the same image will have always the same label (an image with a cat will not be labeled as a cat in the test and as Felis genus in another). Our experiments show that even if there are data quality issues (as in the MIMIC dataset) our method based on this assumption has excellent performance.
>
> Having said this, there are certainly scenarios where our assumption may be violated e.g. when we have features missing not at random (MNAR), or other heterogenous effects. Formalizing and extending our method to such scenarios is a very promising direction for future work.
>
>
>
> > [...] test data for the hospital cannot be fixed because patients are continually coming in.
>
> In this example, the test data consists of only an individual incoming patient's X-ray which requires a prediction. The data-market platform selects the data and trains a model to predict that patient's data and only shares the final model/prediction to the patient. The selected data points are not directly shared. As correctly identified by the reviewer, this means that the procedure needs to be *re-run* for each new test patient.
>
> We stress that only the model/prediction is shared and not the data selected. This is because of data privacy and IP issues. Since data is infinitely and easily copyable, it is not feasible to run a marketplace where raw data is ever shared - the buyer can easily make a copy and resell for a lower price. Thus, in our model, only the trained model/prediction is shared.
>
> We once again thank the reviewer for their great feedback of our work.

---

> > ### Comment · Reviewer_yTne · 2024-08-14
> >
> > Thanks for the detailed rebuttal.

---

### Official Review · Reviewer_MmPt · 2024-07-09

**Soundness:** 3
**Presentation:** 2
**Contribution:** 3
**Rating:** 7
**Confidence:** 4

**Summary:**

This paper uses the linear experimental design to tackle the data acquisition problem.

**Strengths:**

This paper tackles a very challenging but important problem in the data marketplace : evaluating data quality before data transactions as well as minimizing the test error.  The main contribution is the Federated approach for the experimental design.

Motivation is clear and paper is well-organized. The experimental results are convincing.

**Weaknesses:**

1. Overall I like this paper and take a few hours trying to understand the technical part and experimental results. But some necessary analysis and equations are simplified, not so friendly to readers who are not very familiar with the V-optimal experiment design framework and some optimization parts. Please provide some preliminaries in your work.

2. line 136 .. suppose we have a known feature-extractor ...  how to get such good extractor $\phi$ and whether your evaluation results are sensitive to a poor $\phi$?

3. The authors consider the low-budgets cases ( lower 50 training datapoints ) and the validation size for baselines is small. More experiment results are needed.

4. Some parts are confusing for me, please see the questions below.

**Questions:**

1. Could authors explain how to get the eq.(7) derived from the eq.(4) ?

2. $\textbf{confusion part:}$ How $g_i$ in eq.(7)  can be only calculated by the seller $j$ using training data $x_j^{\text{train}}$, given $x_i^{\text{test}}$ is needed( or $\mathbb{E}(x^{\text{test}})$ appears in eq.(7) ) ?

Line 176:  "$g_i$ as well as the update (6) can be trivially computed by seller $j$ using only their data $x_j^{\text{train}}$ (and the test data). "

Line 516:  "our method and Data OOB do not use the validation set"

$\textbf{Question:}$  Whether optimization in eq.(7) involves the test dataset? If it involves, I guess some baselines could also do the same process for data selection : for example,  LAVA could also only use  feature space $\mathcal{X}$ to compare $x^{\text{train}}$ and  $x^{\text{test}}$, and select data points?  If not, back to the question of the confusion part.

 It will be better to provide more details about your analysis or related works for references in the paper for better understanding. ]

3. In the experiment setting, the validation set has only 100 datapoints, could authors conduct the experiments when the validation size increases ?

4. Authors consider low-budgets situations ( 1~10 datapoints ) , it will be better to provide experiments when budgets are larger ?


————————
All my concerns have been addressed. I will raise the score.

---

> ### Author Rebuttal · Authors · 2024-08-05
>
> We thank the reviewer for their comments. We agree that our exposition is dense and some derivations may be inaccessible to reader unfamiliar with experiment design. We will add a more detailed derivation in the Appendix where we will go through all the steps in more detail. Further, we will aim to rewrite in a manner such that despite some of the details being inaccessible, the high-level intuitions and resulting algorithm is accessible to all readers. Having said this, we adress the main concerns raised by the reviewer below.
>
> Also, we would like to stress that the **most useful and challenging setting** for data valuation methods is when the budget is low, and the validation data size is small. This is why we focus on this setting. We believe this is a strength of our work, not a weakness - the other settings are strictly easier. With a large enough budget, even random selection can work. Similarly, if we can collect a large clean validation dataset, the need for a data market is less justified.
>
> > [...] how to get such good extractor $\phi$ and whether your evaluation results are sensitive to a poor $\phi$?
>
> We note that our experiments on Gaussian and MIMIC data are conducted without any feature extractor. In general, it is common to use a pretrained model as a feature extractor but we will provide additional discussion of the choice of feature extractor in the final paper.
>
> > Could authors explain how to get the eq.(7) derived from the eq.(4) ?
>
> Eq 7 is the gradient update for a single datapoint $x_j$. The seller that owns the datapoint $x_j$ can calculate its gradient using the buyer data $x_i^\text{ test }$ and the information matrix $\mathcal{I} := \sum_{j=1}^{n} w_j x_j x_j^\top$.
>
> > "our method and Data OOB do not use the validation set"
>
> We meant that while other methods require a validation set to estimate data value, our approach (and OOB) does not use a held-out validation set to perform data selection.
>
> > Whether optimization in eq.(7) involves the test dataset?
>
> Yes, optimization uses the unlabelled test datapoint $x^\text{ test }_i$. That is, our method is adaptive to the test data point - the train data most relevant to the specific test data is selected. While our focus was to develop a theoretically justified algorithm with provable guarantees, we hope our work will inspire future empirical work that similarly explores selection based on the feature space (including ones based on LAVA). In our current work, we compare against LAVA as described in the paper which requires class labels.
>
> >  In the experiment setting, the validation set has only 100 datapoints, could authors conduct the experiments when the validation size increases ?
>
> On our datasets, we did not find much difference with larger validation sets but we will include more experiments in the revised paper. Also note that as the size of the validation dataset increases, the motivation for trying to select additional training data decreases. Data valuation methods are most useful when the validation dataset is substantially smaller than the training dataset.
>
> > Authors consider low-budgets situations ( 1~10 datapoints ) , it will be better to provide experiments when budgets are larger ?
>
> Our real-world experiments (Fig. 4) use a budget up to 150. We found that past this, the error did not decrease anymore - our method was already filtering out the irrelevant data and selecting the most useful ones. Note that given a high enough budget, even random selection will perform well since eventually all relevant data will get selected. Hence, we decided to focus on the low-budget case which is both the most useful and most challenging. We will add the experiments with higher budgets as well to the appendix.
>
>
> We believe all the concerns raised by the reviewer are addressed above. If so, we again thank the reviewer for their detailed feedback and request them to kindly re-evaluate our work.

---

> > ### Comment · Reviewer_MmPt · 2024-08-09
> >
> > Thanks for your response. All my previous concerns have been addressed. I will raise the score.
> >
> > A quick question, if I try to use the definition of the Sherman-Morrison formula based on
> >
> > $(A+uv^T)^{-1} = A^{-1} - \frac{A^{-1}uv^TA^{-1}}{1+v^TA^{-1}u}$
> >
> > Will the eq(8) should be
> > $
> > P_{t+1}= \frac{1}{1-\alpha_t} P_t - \frac{1}{1-\alpha_t} \times \frac{\alpha_tP_tx_{j_t}x^T_{j_t}P_t}{(1-\alpha_t)+\alpha_tx_{j_t}^TP_tx_{j_t}}
> > $ ?
> >
> > Then is there any guideline to choose an appropriate $\alpha_t$ ?

---

> > > ### Author Response · Authors · 2024-08-09
> > > **Thank you for the pointer**
> > >
> > > This is indeed a typo, thank you for the close reading! Equation (8) should correctly read
> > > $$
> > >  P_{t+1} = \frac{1}{1-\alpha_t}P_t - \frac{\alpha_t P_t x_{j_t} x^\top_{j_t} P_t}{1 - \alpha_t + \alpha_t x_{j_t}^\top P_t x_{j_t}}
> > > $$
> > > Note that the 1 in the denominator in the second term is now $1 - \alpha_t$. We got this by plugging in the following into the Sherman-Morrison formula:
> > > $$
> > > A = \big((1-\alpha_t) \mathcal{I}_t \big)^{-1} = \frac{1}{1-\alpha_t}P_t \quad \text{and } u = \alpha_t x_j \text{ , } v = x_j.
> > > $$
> > >
> > > Our experiments, simulations, and the rest of the code remain unchanged since they used the correct formula (without this typo).

---

> ### Comment · Reviewer_MmPt · 2024-08-10
>
> Thank you. I have checked the code and also run the experiments :) .

---

### Official Review · Reviewer_BRC7 · 2024-07-11

**Soundness:** 4
**Presentation:** 3
**Contribution:** 4
**Rating:** 7
**Confidence:** 5

**Summary:**

This paper introduces a novel method for acquiring training data in a decentralized data market without requiring labeled validation data. Based on detailed and reliable theoretical support, the authors comes to a promising conclusion that the selection methods based on validation set has the performance as bad as the models trained by validation itself. Thus, it theoretically exceeds a series of methods represented by Shapley. This approach is amendable to federated optimization and reduce prediction error, making it more suitable for decentralized markets compared to traditional centralized data valuation methods.

**Strengths:**

S1: This paper combines actual scenarios in the data market to summarize challenges that may exist for sellers and buyers, providing detailed and precise definitions for the data acquisition task. The overall topic is very interesting and practical.
S2: The writing level is very good. Most of the presented viewpoints are well motivated, and the proofs, derivations, and theoretical implications of the formulas mentioned are all well explained and easy to understand.
S3: For data acquisition methods relied on validation data, this paper theoretically deduces the lower bound of its error and proves that validation data(or labels) itself is unnecessary. This is crucial for the field of data acquisition where labels are very insufficient.
S4: The method has good scalability, with sufficient experimental validation.

**Weaknesses:**

W1: The input for data acquisition task is too idealistic. In the actual data market, existing data acquisition tasks are often combined with data discovery, emphasizing the retrieval of data from environments such as data lakes that have different distributions and poor data quality. This paper simplifies the input data as datapoints, lacking consideration for issues related to data quality and failing to explain why datapoints are directly considered without integrating them into the context of the data market.

W2: Although this article provides detailed and solid theoretical deductions, some assumptions are too strong. For example, in Line 127 of Section 4: " First, we assume that the conditional distribution D_(y|x) is identical across Z^Train and Z^Test". In fact, due to the presence of a large number of missing values in the train set, assumptions about data distribution may not hold true. If these assumptions are not valid, the explanation provided in the article regarding V-optimal experiment is insufficient, which makes me confused.

W3: The proof of theorem 1 in the article is interesting, but some details need to be discussed if we want to use this conclusion to overperform methods based on Shapley Value. The existence of parameter σ makes it difficult for me to clearly understand how small this lower bound is. If users have some high-quality validation data (and in practical scenarios of data markets, users should provide high-quality data as much as possible), would this lower bound be acceptable?

W4: The assumption of infinite training data in theorem 1 is not very reasonable. From my experience, the data available in the data market should be a subset of the results from the data discovery task and joinable subsets of datasets in the data lake, with its scale yet to be examined.

W5: The application of kernelized linear regression is not well motivated.

**Questions:**

Q1: Is it reasonable to assume that the training data in theorem1 is infinitely large?
Q2: If the assumptions about data distribution in Line 127 of Section 4(w2) cannot hold true, how does this proof work through V-optimal experiment? Or can you explain why this assumption always holds true even with data quality problems?

**Limitations:**

Yes

---

> ### Author Rebuttal · Authors · 2024-08-05
>
> We greatly thank the reviewer for their close reading of our work, their detailed feedback, and their enthusiastic support. We address the comments raised below.
>
> > W1: The input for data acquisition task is too idealistic ...
>
> We absolutely agree that a truly practical implementation of our method for data markets will involve a lot of very exciting large-scale database research questions including data discovery. As we show, our arguably simplistic model is still very useful to discover important data valuation methods. We hope that our work inspires the database community to integrate data valuation methods such as DAVED to develop real-world data markets.
>
> > W2: ... missing values in the train set, assumptions about data distribution may not hold true...
>
> First, note that some assumptions on the data distribution are necessary. If the train data may be arbitrarily unrelated to the test data distribution, then the only meaningful training data selection strategy is random sampling. Further, our experiments show that when using a high-quality feature extractor, even if there are data quality issues such as missing values (as in the MIMIC dataset) or slight violations of the assumption, our method still has excellent performance.
>
> Having said this, in practice, we imagine that there are instances where our covariate shift assumption is strongly violated - either due to features missing not at random (MNAR) as noted by the reviewer or due to other heterogeneous effects. We believe this represents excellent future research directions which we hope the data valuation community will build upon.
>
>
> > W3: The existence of parameter σ makes it difficult for me to clearly understand how small this lower bound is. If users have some high-quality validation data (and in practical scenarios of data markets, users should provide high-quality data as much as possible), would this lower bound be acceptable?
>
> Let us compare two different strategies users might take. In strategy 1, they will use the validation dataset of $n_{val}$ datapoints to use Shapely value-like methods to select the training data and enjoy some performance worse than $\frac{\sigma^2 d}{n_{val}}$ according to our lower-bound.
>
> In strategy 2, they will completely ignore the training data and simply use the $n_{val}$ validation data to directly train a linear regression model. The error achieved by this strategy is $\frac{\sigma^2 d}{n_{val}}$ (see Proposition 3.5 in [Bach 2024](https://www.di.ens.fr/~fbach/ltfp_book.pdf)). Comparing this to the lower bound above tells us that the user may be better off directly using their validation data to train a model, instead of submitting it to the platform which uses Shapley-value based data selection. We hope this conveys the strength of our lower bound and will include this discussion in the final version of our paper.
>
>
> > W4: The assumption of infinite training data in theorem 1 is not very reasonable.
>
> The assumption that there is infinite training data is only to make the math simpler. We only assume this to guarantee that there is training data of sufficient diversity to select from with high probability. If we had very few training data points, just by random chance, we might end up in a setting where all of them are very similar, making the data selection problem easier. As shown empirically in Fig. 1, 1000 data points are sufficient to satisfy this condition in practice and for prior methods to work comparably to random selection.
>
>
> > W5: The application of kernelized linear regression is not well motivated.
>
> We study the kernelized linear regression setting mostly because it is the only case that is theoretically tractable to analyze. Further, numerous past works have shown that when using a high-quality feature extractor, this kernelized linear proxy model is a good approximation of the full training process (lines 138--143). We also show this empirically in our setting by the performance of our method, and also in Appendix D.5.

---

### Official Review · Reviewer_wuU5 · 2024-07-25

**Soundness:** 3
**Presentation:** 3
**Contribution:** 3
**Rating:** 6
**Confidence:** 4

**Summary:**

The paper introduces a new data valuation algorithm based on linear experimental design. The technique does not require a validation set and can be used in a federated setting.

**Strengths:**

The paper studies a well-motivated trending topic. Data valuation and data selection are important subfields in data-centric ML. I also like the motivation of developing validation-free data valuation algorithms a lot.

**Weaknesses:**

I have reviewed this paper at ICML. I think the new revision has addressed my previous concerns. The following is my ICML review:
- The writing can be improved a bit. For example, in Section 3.3, the proof or citation for the test loss of LS estimator should be provided. In Section 3.4, I don't understand why the constraint of costs can be removed. Such derivation details are better to be explicitly written, it's okey to be in Appendix. I also don't see the benefit of using Frank-Wolfe algorithm.
- The proposed approach requires a feature extractor, and I believe the performance depends greatly on the choice of feature extractor. Afterall, the theoretical justification of this appraoch is based on the assumption that the groundtruth is a noisy linear combination of the feature, that's why the requirement of test labels can be removed. It's good to explicitly acknowledge this point.
- In Figure 3, why does the error significantly increases with full data for many curves? Are there a small portion of the data is being corrupted?

In this new revision, the author has clarified the linear model setup and the derivation is super easy to follow. The idea of using linear model as a proxy is straightforward but I am not aware of other works doing it. My remaining concern is that while this approach does not require a validation set, it requires a feature extractor. My major question is how this feature extraction can be picked in practice. I understand this is a hard question and the use of a public feature embedding is quite common in data selection techniques, therefore I will not reject the paper based on this point.

**Questions:**

See weaknesses.

**Limitations:**

See weaknesses.

---

> ### Author Rebuttal · Authors · 2024-08-05
>
> We thank the reviewer for their comments and the positive evaluation of our work. We agree that the choice of feature extractor is very important. In short, how well our linear proxy model works depends on the quality of our feature extractor. If there are features that are relevant to the task but are not present in the embedding output of the extractor, then the linearity assumption would not hold. Thus, we recommend general-purpose pre-trained foundation models that can extract very broad features from the input. We will add more discussion on the sensitivity of our results to the choice of feature extractors in the final version of our paper.

---

### Decision · Program_Chairs · 2024-09-25

**Decision:**

Accept (poster)

**Comment:**

This paper addresses the problem of selecting valuable data points in a decentralized data marketplace, particularly focusing on domains where data is scarce. The authors propose a federated data acquisition method inspired by linear experimental design, which does not require labeled validation data and instead optimizes data acquisition directly based on test set performance. The method is designed to function efficiently in a decentralized market setting, showing state-of-the-art performance on several real-world medical datasets.

Overall, all reviewers are positive about the paper, appreciating the timeliness of the research question, the novel approach, and the clear writing. There are some reservations, such as the requirement for a high-quality feature extractor, assumptions about data distribution that may not always hold true, and a limited scope regarding budget constraints in practical marketplaces. However, the strengths of the paper clearly outweigh the weaknesses. We recommend accepting the paper and encourage the authors to incorporate the reviewers' comments in their final camera-ready version.